# A Robust Method to Discover Causal or Anticausal Relation

**Yu Yao**[1*], **Yang Zhou**[1*], **Bo Han**[2,3], **Mingming Gong**[4,6], **Kun Zhang**[5,6], **Tongliang Liu**[1,6†]

[1]Sydney AI Centre, The University of Sydney, [2]TMLR Group, Hong Kong Baptist University
[3]RIKEN Center for Advanced Intelligence Project, [4]The University of Melbourne
[5]Carnegie Mellon University, [6]Mohamed bin Zayed University of Artificial Intelligence

## Abstract

Understanding whether the data generative process follows causal or anticausal relations is important for many applications. Existing causal discovery methods struggle with high-dimensional perceptual data such as images. Moreover, they require well-labeled data, which may not be feasible due to measurement error. In this paper, we propose a robust method to detect whether the data generative process is causal or anticausal. To determine the causal or anticausal relation, we identify an asymmetric property: under the causal relation, the instance distribution does not contain information about the noisy class-posterior distribution. We also propose a practical method to verify this via a noise injection approach. Our method is robust to label errors and is designed to handle both large-scale and high-dimensional datasets effectively. Both theoretical analyses and empirical results on a variety of datasets demonstrate the effectiveness of our proposed method in determining the causal or anticausal direction of the data generative process.

## 1 Introduction

In a dataset containing feature variables and a class variable, the dataset is considered to have a *causal* relation if some feature variables cause the class variable, but the class variable does not cause any feature variables. Conversely, the dataset is considered to have an *anticausal* relation if the class variable causes some feature variables.

Understanding whether a dataset follows causal or anticausal relations is crucial for strategic decision-making across different domains. In semi-supervised learning (SSL), correctly identifying causal and anticausal relations helps determine whether SSL methods should be used to improve predictions (Kügelgen et al., 2020; Huang et al., 2021). In transfer learning, understanding these relations reveals distribution shifts and guides the selection of appropriate transfer strategies (Schölkopf et al., 2012; Huang et al., 2023). Additionally, there are many other potential applications in causal discovery (Peters et al., 2017b; Zanga et al., 2022). For instance, in healthcare, one may want to determine whether lifestyle factors (such as exercise duration or dietary habits, which are continuous variables) lead to specific health outcomes (like the development of diabetes or heart disease, which are discrete variables). In environmental science, researchers might be interested in whether environmental conditions (such as the pollution index, which is a continuous variable) cause specific ecological events (like the occurrence of acid rain, which is a discrete variable), or if the relation is reversed.

However, in real-world applications, it is often unclear whether a dataset is causal or anticausal. Existing causal discovery methods face several challenges (*For detailed related work, please refer to Appendix. B*). The first challenge arises when dealing with datasets consisting of perceptual data, such as images or audio. In these situations, feature variables such as orientation and lighting conditions, which are hidden behind the images, are unobservable (Schölkopf et al., 2021). Most existing causal discovery methods are designed to detect relations between observed feature variables (Kalainathan et al., 2020; Shimizu et al., 2011; Huang et al., 2018; Geiger & Heckerman, 1994; Zhang & Hyvarinen, 2009; Peters et al., 2011; 2014; Chen & Chan, 2013), making them ill-suited

---

†Correspondence to Tongliang Liu (tongliang.liu@sydney.edu.au).
*These authors contributed equally.

for these types of datasets. Currently, we are unaware of any method that can effectively determine causal or anticausal relations in such datasets.

Additionally, in real-world scenarios, observed labels in large-scale datasets often contain errors (Deng et al., 2009; Xiao et al., 2015; Yuan et al., 2023), which have not been considered by existing causal discovery methods. In the mining process of large-scale datasets, inexpensive but imperfect annotation methods are widely employed, such as querying commercial search engines (Li et al., 2017), downloading social media images with tags (Mahajan et al., 2018), or leveraging machine-generated labels (Kuznetsova et al., 2020). These methods inevitably yield examples with label errors. When label errors are present, the randomness of these errors affects the strength of the causal dependence between features and the observed (noisy) label $\tilde{Y}$, making it more challenging to discern the true relations accurately. Existing methods often use conditional independence tests (Zhang & Hyvarinen, 2009; Peters et al., 2011; 2014; Sun et al., 2025) or score optimizations (Imoto et al., 2002; Hyvärinen & Smith, 2013; Huang et al., 2018) to evaluate the strength and structure of these relations. Label errors introduce random fluctuations that distort the underlying relations between features and labels. Consequently, these tests or score optimizations may be misled by the noise, leading to inaccurate estimations of the relations.

In this paper, we introduce a robust method aimed at determining whether a dataset is causal or anticausal. We found that even when data contain label errors, it is possible to leverage observed labels $\tilde{Y}$ as they contain information about clean classes. Specifically, let $X$ and $\tilde{Y}$ denote the instance (e.g., an image) and the observed label, respectively. There is an asymmetric property: *under anticausal datasets, the distribution of instances $P(\mathbf{X})$ can help predict observed labels $\tilde{Y}$, but this does not hold under causal datasets*. We designed a practical estimator to check this property.

Intuitively, we check whether the distribution of the instance $P(\boldsymbol{X})$ can help predict the observed label $\tilde{Y}$. To achieve it, we generate pseudo labels using unsupervised methods (Van Gansbeke et al., 2020; Ghosh & Lan, 2021)[*]. *In a causal dataset*, the distribution $P(\boldsymbol{X})$ does not contain useful information for predicting the observed label $\tilde{Y}$. Therefore, injecting different levels of label noise into the observed labels does not affect the (average) disagreement between pseudo labels and the observed labels. Conversely, *in an anticausal dataset*, the distribution $P(\boldsymbol{X})$ contains useful information for predicting the observed label $\tilde{Y}$. Injecting noise in this case introduces randomness, which makes the observed labels less predictable as the noise level increases, thus changing the disagreement between pseudo labels and observed labels.

In Section 3.3, we theoretically prove that in a causal dataset, the disagreement between pseudo labels and observed labels remains unchanged with varying noise levels. In contrast, in an anticausal setting, the disagreement changes as the noise levels change. It is also worth noting that our RoCA estimator is general and can handle different types of label errors defined in existing literature, including random classification label errors (Wang et al., 2019), asymmetric label errors (Scott et al., 2013), manifold label errors (Cheng et al., 2022), and part-dependent label errors (Xia et al., 2020). Experimental results on different datasets demonstrate that our method can accurately determine whether a dataset is causal or anticausal.

## 2 PRELIMINARIES

Let $D$ be the distribution of a pair of random variables $(\boldsymbol{X}, \tilde{Y}) \in \mathcal{X} \times \{1, \ldots, C\}$, where $C$ denotes the number of classes, $\boldsymbol{X}$ represents an instance, and $\tilde{Y}$ denotes observed label, which may not be identical to the clean class $Y$. Given a training sample $\boldsymbol{S} = \{\boldsymbol{x}_i, \tilde{y}_i\}_{i=1}^m$, we aim to reveal whether the dataset is causal or anticausal.

**The Principle of Independent Mechanisms** According to independent mechanisms (Peters et al., 2017b), the causal generative process of a system's variables consists of autonomous modules. Crucially, these modules neither inform nor influence each other. In the probabilistic cases detailed in Chapter 2 of Peters et al. (2017b), the principle states that "*the conditional distribution of each variable given its causes (i.e., its mechanism)*

---

[*]Specifically, clusters are formed based on the distribution of instances $P(\boldsymbol{X})$, and a pseudo label $Y'$ is assigned to each cluster based on the majority of observed labels within that cluster

*does not inform or influence the other conditional distributions.*" In other words, assuming all underlying causal variables are given, the conditional distributions of each variable do not share information and are independent of each other given all its causal parents (which can be an empty set).

**Causal or Anticausal** We follow the definition of Causal and Anticausal datasets from Schölkopf et al. (2012). For causal datasets, some variables in $\boldsymbol{X}$ act as causes of the class $Y$, and no variable in $\boldsymbol{X}$ is an effect of the class $Y$ or shares a common cause with the class $Y$ (e.g., Fig. 1a). In this case, $Y$ can only be an effect of some variables in $\boldsymbol{X}$. Two distributions $P(\boldsymbol{X})$ and $P(Y|\boldsymbol{X})$ satisfy the independent causal mechanisms. The distribution $P(\boldsymbol{X})$ does not contain information about $P(Y|\boldsymbol{X})$. For anticausal datasets, the class $Y$ causes some of the variables in $\boldsymbol{X}$ (e.g., Fig. 1b). In this case, the independent causal mechanisms are not satisfied for $P(\boldsymbol{X})$ and $P(Y|\boldsymbol{X})$, implying that $P(\boldsymbol{X})$ contains information about $P(Y|\boldsymbol{X})$ or that $P(\boldsymbol{X})$ can help predict class $Y$, intuitively.

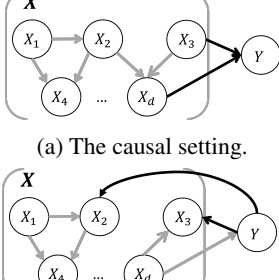

(a) The causal setting.

(b) The anticausal setting.

Figure 1: The direction of the black edge determines whether a dataset is causal or anticausal.

## 3 A ROBUST CAUSAL AND ANTICAUSAL ESTIMATOR

In this section, we present a practical and **Ro**bust **C**ausal and **A**nticausal (RoCA) Estimator designed to infer whether a dataset is causal or anticausal while taking into account the presence of label errors in observed labels. Note that the assumption of our method discussed in Appendix G

### 3.1 RATIONALE BEHIND ROCA

**Data Generative Processes with Label Errors** *A dataset with label errors can be viewed as a result of a random process where labels are flipped based on certain probabilities*. Data generation involves two stages (see Fig. 2). Initially, an annotator is trained using a clean set $Z$, acquiring specific prior knowledge, $\theta$, for the labeling task. This knowledge helps the annotator form an annotation mechanism $P_\theta(\tilde{Y}|\boldsymbol{X})$, approximating the true class posterior $P(Y|\boldsymbol{X})$. This mechanism, being correlated with $P(Y|\boldsymbol{X})$, provides insights into the true class posterior. In the annotation phase, the annotator encounters a new instance $\boldsymbol{X}$ without an observed clean class $Y$. Using the prior

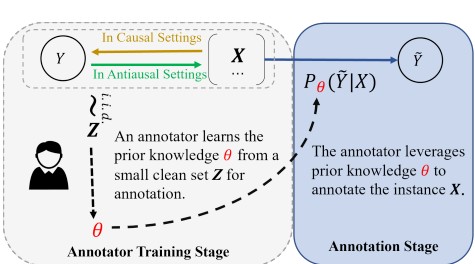

Figure 2: Labeling errors in annotations.

knowledge $\theta$, the annotator assigns an observed label $\tilde{Y}$ based on $P_\theta(\tilde{Y}|\boldsymbol{X})$. This process can sometimes lead to mislabeling. It's noteworthy that $P_\theta(\tilde{Y}|\boldsymbol{X})$ *generally maintains a dependence with* $P_\theta(Y|\boldsymbol{X})$. Imagine if this dependence did not exist; the annotation mechanism $P_\theta(\tilde{Y}|\boldsymbol{X})$ would essentially be a random guess of $P(Y|\boldsymbol{X})$, rendering the observed label $\tilde{Y}$ meaningless. We will demonstrate that, due to this dependence, $P_\theta(\tilde{Y}|\boldsymbol{X})$ can serve as a surrogate for $P(Y|\boldsymbol{X})$ to help determine whether a dataset is causal or anticausal.

**Overview** In line with the principle of independent mechanisms (Peters et al., 2017b), to determine causal and anticausal relationships, one can check whether $P(\boldsymbol{X})$ can help predict $P(Y|\boldsymbol{X})$ or not. If it does, the relationship is anticausal; otherwise, it is causal. However, in the presence of label errors, the clean class $Y$ becomes latent. Instead, we have observed labels $\tilde{Y}$ containing errors. In this case, instead of checking whether $P(\boldsymbol{X})$ contains information about $P(Y|\boldsymbol{X})$, our method checks if $P(\boldsymbol{X})$ contains information about $P_\theta(\tilde{Y}|\boldsymbol{X})$. Since $P(\tilde{Y}|X)$ can be regarded as an estimation of $P(Y|\boldsymbol{X})$ with errors. If $P(\boldsymbol{X})$ can help predict the posterior of observed labels $P(\tilde{Y}|\boldsymbol{X})$, it can also help in predicting the posterior of classes $P(Y|\boldsymbol{X})$.

$P_\theta(\tilde{Y}|\boldsymbol{X})$ **Serves as a Surrogate of** $P(Y|\boldsymbol{X})$   Reminding that for causal datasets, to determine whether a dataset is causal or anticausal, one can examine whether $P(\boldsymbol{X})$ can inform $P(Y|\boldsymbol{X})$. Specifically, according to the independent mechanisms (Kügelgen et al., 2020; Peters et al., 2017b), on causal datasets, $P(\boldsymbol{X})$ does not provide any information about $P(Y|\boldsymbol{X})$; on anticausal datasets, $P(\boldsymbol{X})$ generally contains information about $P(Y|\boldsymbol{X})$. However, when data contains label errors, the clean label $Y$ is latent, estimating $P(Y|\boldsymbol{X})$ is challenging. *One natural thought is to find a surrogate distribution that can help in determining the causal direction.* Specifically, the surrogate distribution should satisfy an asymmetric property with two key conditions. *1). In a causal setting, $P(\boldsymbol{X})$ should not contain information about the surrogate distribution; 2). In an anticausal setting, $P(\boldsymbol{X})$ should contain information about the surrogate distribution.*

If such a surrogate can be found, we can infer whether a dataset is causal or anticausal by examining whether $P(\boldsymbol{X})$ contains information about the surrogate distribution. We find that $P_\theta(\tilde{Y}|\boldsymbol{X})$ fits these requirements. As it is an approximation to the underlying distribution $P(Y|\boldsymbol{X})$. It statically depends on and contains information about $P(Y|\boldsymbol{X})$. Moreover, under a causal setting, $P(\boldsymbol{X})$ cannot inform $P_\theta(\tilde{Y}|\boldsymbol{X})$, since $\tilde{Y}$ and $Y$ are effects of $\boldsymbol{X}$, and $P(\boldsymbol{X})$ and $P_\theta(\tilde{Y}|\boldsymbol{X})$ follow causal factorization and are independent according to independent mechanisms (Peters et al., 2017b). Thus, $P_\theta(\tilde{Y}|\boldsymbol{X})$ is a proper surrogate.

**Validating Whether** $P(\boldsymbol{X})$ **Contains Information About** $P_\theta(\tilde{Y}|\boldsymbol{X})$   Building on the analysis that $P_\theta(\tilde{Y}|\mathbf{X})$ can serve as a surrogate for $P(Y|\mathbf{X})$, the remaining challenge is to effectively infer whether $P(\mathbf{X})$ contains information about $P_\theta(\tilde{Y}|\mathbf{X})$. In other words, we need to determine *whether some information learned from $P(\mathbf{X})$ can be leveraged to help predict $P_\theta(\tilde{Y}|\mathbf{X})$*. Intuitively, to achieve this, our proposed estimator employs unsupervised or self-supervised algorithms on $P(\boldsymbol{X})$ to generate clusters. We then assign a pseudo label $Y'$ to each cluster based on the majority of observed labels within it. If these pseudo labels are informative to observed labels, it indicates that $P(\boldsymbol{X})$ contains information about $P_\theta(\tilde{Y}|\boldsymbol{X})$.

To identify regions that can help predict observed (noisy) labels, different levels of noise are manually injected into observed labels. To check if these pseudo labels are informative to observed labels, we need to validate whether each pseudo label $Y'$ is a random guess of its corresponding observed label $\tilde{Y}$ given an instance $\boldsymbol{X}$. Specifically, let $C$ be the number of classes, *the asymmetric property becomes: 1). In a causal setting, $P(\tilde{Y} = \tilde{y}|Y' = y', \boldsymbol{X} = \boldsymbol{x}) = 1/C$ for each instance; 2). In an anticausal setting, $P(\tilde{Y} = \tilde{y}|Y' = y', \boldsymbol{X} = \boldsymbol{x}) \neq 1/C$ for some instances.*

However, accurately estimating the distribution $P(\tilde{Y}|Y', \boldsymbol{X})$ from data can be challenging. Firstly, the feature vector or instance $\boldsymbol{X}$ can be high-dimensional, making the estimation of the distribution difficult due to the curse of dimensionality (Köppen, 2000). As the dimensionality increases, the data becomes sparse, requiring an exponentially larger amount of data to maintain estimation accuracy. Moreover, it is difficult to have prior knowledge or make parametric assumptions about the distribution $P(\tilde{Y}|Y', \boldsymbol{X})$, which increases the difficulty of estimation.

**Avoiding Estimation of** $P(\tilde{Y}|Y', \boldsymbol{X})$ **via Noise Injection**   To avoid directly estimating $P(\tilde{Y}|Y', \boldsymbol{X})$, we propose a simple and effective noise-injection method. We found that we can inject different levels of instance-dependent noise to the observed label $\tilde{Y}$, then compare the trend of the average disagreement between pseudo labels and modified labels under different levels of noise. The rationale is that, under the causal setting, $P(\boldsymbol{X})$ does not contain information about the surrogate distribution $P_\theta(\tilde{Y}|\boldsymbol{X})$. Therefore, exploiting $P(\boldsymbol{X})$ cannot help predict observed labels. As a result, the pseudo labels obtained from $P(\boldsymbol{X})$ are random guesses of the observed labels. If we introduce noise to these observed labels by randomly flipping some of them, the pseudo labels should continue to guess the modified labels randomly. Since pseudo labels randomly guess any label with a fixed probability of $1/C$, the average disagreement between pseudo labels and the modified labels remains consistent, regardless of the noise level.

By contrast, in the anticausal setting, $P(\boldsymbol{X})$ contains information about the surrogate distribution $P_\theta(\tilde{Y}|\mathbf{X})$. This implies that pseudo labels, when derived by sufficiently exploiting $P(\boldsymbol{X})$, are not random guesses of the observed labels in general. As we progressively modify the observed labels by injecting increasing levels of noise, these modified labels become more random and unpredictable.

Figure 3: A toy example of the change of the label disagreement via our noise injection.

This shift results in a change in the level of disagreement between the pseudo labels and the modified labels. In the end, our RoCA estimator validates the asymmetric property by examining ***whether there is a change in the average disagreement between pseudo labels (obtained through unsupervised methods) and modified labels (derived by injecting observed labels with increasing levels of noise)***.

**A Toy Example** To provide more intuition about the noise injection, let's consider two toy binary classification datasets illustrated in Fig. 3, where the instance $\boldsymbol{X} \in \mathbb{R}^2$. Assume that an unsupervised method separates instances into two clusters, with half of them assigned the pseudo label $Y' = 0$, and the other half assigned $Y' = 1$. We'll focus on instances with the pseudo label $Y' = 1$, which are located in two regions ($R_1$ and $R_2$) based on their $\boldsymbol{X}$ values.

In Fig. 3 (a), on a causal dataset, before noise injection, the distribution of observed labels in regions $R_1$ and $R_2$ indicates that $P(\tilde{Y} = 1|Y' = 0, \boldsymbol{X} = x) = P(\tilde{Y} = 0|Y' = 0, \boldsymbol{X} = x) = 1/2$. This suggests that each instance's pseudo label is a random guess of its observed label, rendering an average disagreement $P(\tilde{Y}^\rho = 1|Y' = 0)$ as $1/2$. After noise injection, say with an instance-dependent noise flipping $40\%$ and $20\%$ observed labels in regions $R_1$ and $R_2$, the average disagreement remains the same. It indicates no trend in average disagreements between pseudo labels and modified labels across different noise levels.

Fig. 3 (b) demonstrates an anticausal dataset scenario. *Despite the average disagreement for the observed label $\tilde{Y} = 0$ being $0.5$, each instance's pseudo label isn't a random guess of its observed label.* Since in region $R_1$, all instances have the observed label $\tilde{Y} = 0$, it implies that $R_1$ in $P(\boldsymbol{X})$ contains information about predicting the observed label $\tilde{Y} = 0$. Similarly $R_2$ in $P(\boldsymbol{X})$ contains information about predicting the observed label $\tilde{Y} = 1$. This results in $P(\tilde{Y} = 1|Y' = 0, \boldsymbol{X} = x) = 1$ in region $R_1$ and $P(\tilde{Y} = 0|Y' = 0, \boldsymbol{X} = x) = 1$ in region $R_2$, deviating from the expected $1/2$. After injecting the same instance-dependent noise into observed labels in regions $R_1$ and $R_2$, the average disagreement $P(\tilde{Y}^\rho = 1|Y' = 0)$ drops to $0.3$, reflecting the regions where $P(\tilde{Y} = \tilde{y}|Y' = y', \boldsymbol{X} = x)$ doesn't equal $1/C$. Thus, a trend in the average disagreements can be found under different noise levels.

## 3.2 IMPLEMENTATION OF RoCA ESTIMATOR

The core idea of our method is to check if the distribution of instances $P(\boldsymbol{X})$ carries relevant information about the prediction task $P(\tilde{Y}|\boldsymbol{X})$ to determine whether a dataset is causal or anticausal. To achieve it, we generate clusters by employing advanced unsupervised methods (Van Gansbeke et al., 2020; Ghosh & Lan, 2021). Then a pseudo label $Y'$ is assigned to each cluster based on the majority of observed labels within the cluster. To identify regions that can help predict observed (noisy) labels, different levels of noise are manually injected into observed labels. By using 0-1 loss, we calculate the average disagreements between pseudo labels and the modified labels with different injected noise levels, respectively. In a causal setting, the average disagreements remain the same under different noise levels; in an anticausal setting, the disagreement and the noise level are dependent, and a trend can be found.

**Learning Pseudo Labels** To learn pseudo labels, firstly, instances are clustered using a chosen unsupervised algorithm. Then each cluster is then assigned a pseudo label $Y'$ based on the majority of observed labels within that cluster. Since our objective is to check if pseudo labels are random guesses of observed labels. Therefore the cluster number is set to be identical to the number of observed labels. More specifically, consider $K = i$ as the $i$-th cluster ID, and $\boldsymbol{X}_{K=i}$ as the set of

instances with the $i$-th cluster ID, i.e.,

$$\boldsymbol{X}_{K=i} = \{\boldsymbol{x}|(\boldsymbol{x}, \tilde{y}) \in S, f(\boldsymbol{x}) = i\},$$

where $f$ is a clustering algorithm that assigns an instance $\boldsymbol{x}$ with a cluster ID. Similarly, let $\boldsymbol{X}_{\tilde{Y}=j}$ denote the set of instances with the observed label $\tilde{Y} = j$. Let $\mathbb{1}_A$ be an indicator function that returns 1 if the event $A$ holds true and 0 otherwise. The pseudo label $Y'$ assigned to the instances in the set $\boldsymbol{X}_{k=i}$ is determined by applying the Hungarian assignment algorithm (Jonker & Volgenant, 1986) which ensures an optimal assignment of pseudo labels to clusters such that the total number of mislabeled instances within each cluster is minimized.

**An Instance-Dependent Noise Injection**    Our method is based on noise injection. Firstly, *the injected label noise should depend on the instance*. For example, as illustrated in Fig. 3, the noise rates of instances in different regions $R_1$ and $R_2$ are different. When noise rates for different instances are different, the disagreement between pseudo labels and modified labels changes after the noise injection on the anticausal dataset. Moreover, according to Theorem 1, in causal settings, to make the disagreement between pseudo labels and modified labels remain consistent across different noise levels, the noise must be designed in a particular way. Specifically, *the design of the label-noise distribution has to fulfill that the probability of flipping an observed label to any other class is uniformly distributed*, i.e.,

$$P(\tilde{Y}^\rho = i|\tilde{Y} = j, x) = \frac{\rho_x}{C-1} \text{ for all } i \neq j, \tag{1}$$

where $\rho_x = P(\tilde{Y}^\rho \neq \tilde{Y}|X = x)$ represents the flip rate of an instance $\boldsymbol{x}$. Furthermore, $\rho = \mathbb{E}_{\boldsymbol{X}}[\rho_x]$ represents the expected noise level injected into the dataset. The notation $\tilde{Y}^\rho$ refers to the modified label after injecting $\rho$-level label noise.

We implement a type of label noise that meets the aforementioned conditions. To let the designed label noise be instance dependent, we determine the flip rate magnitude based on the $\ell_1$ norm of the instance $\boldsymbol{X}$. Subsequently, we set the flip rate to be uniformly distributed across other classes to fulfill Eq. (1). Specifically, for each instance in the dataset, we compute its $\ell_1$ norm. These computed norms are stored in a vector $\boldsymbol{A}$. Subsequently, we generate a vector $\boldsymbol{P}$ of length $m$, where each element represents a flip rate sampled from a truncated normal distribution $\psi$. The mean of the truncated normal distribution is the expected noise level to be injected, the variance is set to 1, the lower limit is 0, and the upper limit is 1. To make the label noise depend on instances, we build dependence between the instances and the sampled flip rates in $\boldsymbol{P}$ by sorting both $\boldsymbol{A}$ and $\boldsymbol{P}$ in ascending order. As a result, an instance with a smaller $\ell_1$ norm $a_i$ in $\boldsymbol{A}$ is associated with a lower individual flip rate $\rho_i$ in $\boldsymbol{P}$. *The pseudocode for our noise generation is provided in Appendix E.*

Note that the mean value of a truncated Gaussian distribution does not reflect the actual noise level. For example, if its mean value is set to 0.2 and truncated at 0, the actual expected noise level will be smaller than 0.2. Therefore, in our calculation of the regression coefficient for disagreement under different noise levels, we use the actual noise level computed directly from the data rather than relying on the mean value of the truncated distribution.

**Measuring the Change of the Average Disagreement via the Regression Coefficient**    To infer whether a dataset is causal or anticausal, the key is to determine whether the average disagreement between pseudo labels (obtained through unsupervised methods) and modified labels (derived by injecting observed labels with increasing levels of noise) changes. In causal scenarios, there should be no change, whereas in anticausal settings, a change is expected. Therefore, we inject different levels of label noise and measure the trend of disagreements as the level of injected label noise increases by employing a regression model.

To measure the disagreement, let $\boldsymbol{Y}'$ be the set containing pseudo labels for $\{\boldsymbol{x}_1, \boldsymbol{x}_2, \ldots, \boldsymbol{x}_m\}$. Let $\tilde{\boldsymbol{Y}}^\rho$ be the set of modified labels for $\{\boldsymbol{x}_1, \boldsymbol{x}_2, \ldots, \boldsymbol{x}_m\}$ after injecting instance-dependent noise with an expected noise level $\rho$. The disagreement between the pseudo-label set $\boldsymbol{Y}'$ and the modified label set $\tilde{\boldsymbol{Y}}^\rho$ is measured using the 0-1 loss, $\ell_{01} = \frac{\sum_{i=1}^m \mathbb{1}_{\{y'_i \neq \tilde{y}_i^\rho\}}}{m}$.

To measure the trend of the disagreement, we employ a linear regression model. Specifically, we *uniformly* sample 20 average noise levels from 0 to 0.5. We then inject label noise with each sampled

averaged noise level (denoted as $\rho^i$) into the observed labels and calculate the average disagreement using the 0-1 loss. As a result, for each noise level $\rho^i$, a corresponding disagreement is calculated. Next, the linear regression model is employed to characterize the dependence between the noise level $\rho$ and the loss $\ell_{01}$. The objective is as follows.

$$\{\hat{\beta}_0, \hat{\beta}_1\} = \arg\min_{\beta_0, \beta_1} \frac{1}{n} \sum_{i=1}^{n} (\ell_{01}^i - (\beta_1 \rho^i + \beta_0))^2, \tag{2}$$

where $\hat{\beta}_0$, $\hat{\beta}_1$ refer to the estimated intercept and regression coefficient of the regression line, $\ell_{01}^i$ denotes $0-1$ loss calculated under the $\rho^i$ noise level, respectively, and $n$ is the total number of sampled noise levels. Accordingly, for causal datasets, the regression coefficient $\hat{\beta}_1$ should approximate 0. In contrast, for anticausal datasets, this regression coefficient should deviate significantly from 0.

### 3.3 THEORETICAL ANALYSES

We theoretically show that RoCA estimator holds the aforementioned asymmetric property; therefore, it can detect causal and anticausal direction. Specifically, by applying RoCA estimator, under the causal setting, the disagreement and the noise level should not be dependent on each other, i.e., the regression coefficient $\beta_1$ is 0 in Theorem 1; under the anticausal setting, the disagreement and the noise level are dependent on each other, i.e., the regression coefficient $\beta_1$ is not 0 in Theorem 2.

Let $\mathcal{X}$ be the instance space and $C$ the set of all possible classes. Let $S = \{(x_i, \tilde{y}_i)\}_{t=0}^{m}$ be a sample set. Let $h : \mathcal{X} \rightarrow \{1, \ldots, C\}$, be a hypothesis that predicts pseudo labels of instances. Concretely, it can be a K-means algorithm together with the Hungarian algorithm which matches the cluster ID to the corresponding pseudo labels. Let $\mathcal{H}$ be the hypothesis space, where $h \in \mathcal{H}$. Let $\tilde{R}^\rho(h) = \mathbb{E}_{(\boldsymbol{x}, \tilde{y}^\rho) \sim P(\boldsymbol{X}, \tilde{Y}^\rho)}[\mathbb{1}_{\{h(\boldsymbol{x}) \neq \tilde{y}^\rho\}}]$ be the expected disagreement $\tilde{R}(h)$ between pseudo labels and generated labels $\tilde{y}^\rho$ with $\rho$-level noise injection. Let $\hat{\tilde{R}}_S^\rho(h)$ be the average disagreement (or empirical risk) of $h$ on the set $S$ after $\rho$-level noise injection. Theorem 1 and Theorem 2 leverage the concept of *empirical Rademacher complexity*, denoted as $\hat{\mathfrak{R}}_S(\mathcal{H})$ (Mohri et al., 2018).

**Theorem 1** (Invariant Disagreements Under the Causal Settings). *Under the causal setting, assume that for every instance and clean class pair $(x, y)$, its observed label $\tilde{y}$ is obtained by a noise rate $\rho_x$ such that $P(\tilde{Y} = \tilde{y}|Y = y, \boldsymbol{X} = x) = \frac{\rho_x}{C-1}$ for all $\tilde{y} \neq y \wedge \tilde{y} \in C$. Then after injecting noise to the sample with arbitrary average noise rates $\rho^1$ and $\rho^2$ such that $0 \leq \rho^1 \leq \rho^2 \leq 1$, with a $1 - \delta$ probability and $\delta > 0$,*

$$|\hat{\tilde{R}}_S^{\rho^1}(h) - \hat{\tilde{R}}_S^{\rho^2}(h)| \leq 4\hat{\mathfrak{R}}_S(\mathcal{H}) + 6\sqrt{\frac{\log\frac{4}{\delta}}{2m}}. \tag{3}$$

As the sample size $m$ increases, the term $3\sqrt{\frac{\log\frac{4}{\delta}}{2m}}$ tends towards 0 at a rate of $\mathcal{O}(\frac{1}{\sqrt{m}})$. Additionally, the empirical Rademacher complexity $\hat{\mathfrak{R}}S(\mathcal{H})$ of the K-means algorithm also tends towards 0 at a rate of $\mathcal{O}(\frac{1}{\sqrt{m}})$, as demonstrated by Li & Liu (2021). Consequently, the right-hand side of Inequality (3) converges to 0 at a rate of $\mathcal{O}(\frac{1}{\sqrt{m}})$. This implies that as the sample size increases, the difference between the disagreements $\hat{\tilde{R}}_S^{\rho^1}(h)$ and $\hat{\tilde{R}}_S^{\rho^2}(h)$, obtained by introducing different noise levels, will tend to 0. In other words, the level of disagreement remains unaffected by changes in noise levels, consequently leading to the conclusion that the regression coefficient $\beta_1$ equals zero.

**Theorem 2** (Variable Disagreements Under the Anticausal Setting). *Under the anticausal setting, after injecting noise with a noise level $\rho = \mathbb{E}_X[\rho_x]$, $\tilde{R}^\rho(h) - \tilde{R}(h) = \mathbb{E}\left[\left(1 - \frac{C\tilde{R}(h,x)}{C-1}\right)\rho_x\right]$.*

Theorem 2 shows that the difference in disagreements after noise injection, compared to disagreements on observed labels, is $\mathbb{E}\left[\left(1 - \frac{C\tilde{R}(h,x)}{C-1}\right)\rho_x\right]$. Under the anticausal setting, the pseudo labels predicted by $h$ are not random guesses. In this case, $\tilde{R}(h,x) \neq (C-1)/C$, then the difference is always nonzero. It implies that after injecting noise, the regression coefficient $\beta_1$ will be nonzero.

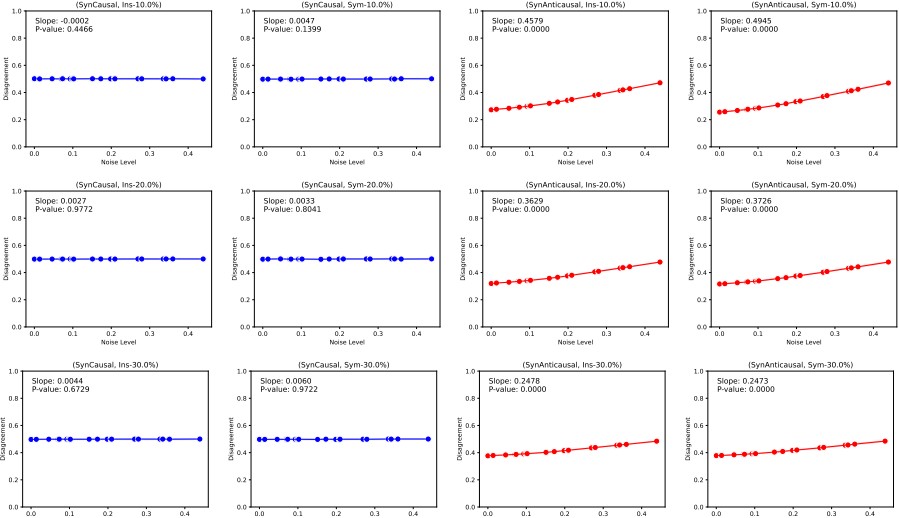

Figure 4: The change in the (average) disagreements and their standard deviations (which are minimal) between pseudo labels $Y'$ and modified labels $\tilde{Y}^\rho$ with the increase in the noise level $\rho$ for the synthetic datasets *synCausal* and *synAnticausal*.

## 4 EXPERIMENTS

We evaluate RoCA estimator across 22 datasets. This includes 2 synthetic datasets (*synCausal* and *synAnticausal*), 3 bivariate datasets (for checking pairwise causal relations with only two variables), 14 multivariate datasets, and 3 image datasets (*CIFAR10* (Krizhevsky et al., 2009), *CIFAR10N* (Wei et al., 2022), and *Clothing1M* (Xiao et al., 2015)). Notably, *CIFAR10N* contains 5 real-world label-error settings named "Wors", "Aggre", "Random1", "Random2", and "Random3". *Clothing1M* also contains real-world label errors and is a large-scale dataset with 1M images. We use K-means clustering method (Likas et al., 2003) for non-image datasets and the SPICE* clustering method (Niu et al., 2021) for image datasets to obtain the pseudo labels $Y'$. Our objective is to check if pseudo labels are random guesses of observed labels, so the number of unique pseudo labels matches the number of unique observed labels. Therefore, the cluster number is set to be identical to the number of unique observed labels.

To validate the robustness of RoCA estimator, different label errors are employed on synthetic datasets and non-image datasets: 1. **Symmetry Flipping** (Sym) (Patrini et al., 2017), which randomly replaces a percentage of labels in the training data with all possible labels. 2. **Pair Flipping** (Pair) (Han et al., 2018), where labels are only replaced by similar classes. For datasets with binary class labels, Sym and Pair noises are identical. 3. **Instance-Dependent Label Errors** (IDN) (Xia et al., 2020), where different instances have different transition matrices depending on parts of instances. To simulate scenarios with label errors, different errors are injected into the clean classes.

Different norms were used to generate instance-dependent label noise. For the UCI datasets, a norm is directly applied to the covariates, which represent meaningful features of each instance. For image datasets, where raw pixel values do not capture meaningful semantic features, we employ a Variational Autoencoder (VAE) to extract feature representations that encapsulate the semantic meaning of each instance. The norm is applied to these feature representations rather than the raw pixel values. In this setup, the latent dimension of the VAE is set to 10, and a ResNet-18 is used as the backbone for feature extraction. Note that to further enhance the dependence of label noise on semantic features, iVAE (Karmeshu & Jain, 2003) can also be used. Empirically, using iVAE and VAE on the above datasets yields almost similar results.

### 4.1 EXPERIMENTS ON SYNTHETIC DATASETS

We generated two additional synthetic datasets, *synCausal* and *synAnticausal*, to validate our RoCA estimator. Each dataset consists of 20,000 instances with five attributes and one label. In the case of *synCausal*, we generate each instance by randomly sampling 5 values from a standard normal distribution to represent $\boldsymbol{X}$, and then compute the corresponding $Y$ value using a polynomial function.

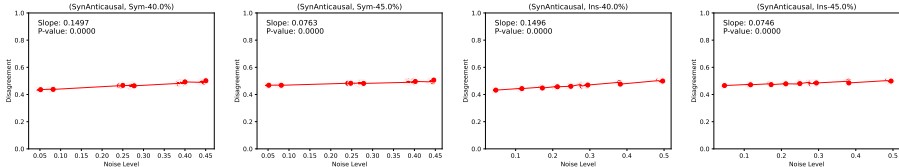

Figure 5: Performance of RoCA on the synAnticausal dataset when $X$ and $\tilde{Y}$ have weak dependence

| SynCausal | | | | | | | |
|---|---|---|---|---|---|---|---|
| | 0% | Ins-10% | Ins-20% | Ins-30% | Sym-10% | Sym-20% | Sym-30% |
| $l_0$ | causal | causal | causal | causal | causal | causal | causal |
| $l_2$ | causal | causal | causal | causal | causal | causal | causal |
| $l_\infty$ | causal | causal | causal | causal | causal | causal | causal |

| SynAnticausal | | | | | | | |
|---|---|---|---|---|---|---|---|
| | 0% | Ins-10% | Ins-20% | Ins-30% | Sym-10% | Sym-20% | Sym-30% |
| $l_0$ | anticausal | anticausal | anticausal | anticausal | anticausal | anticausal | anticausal |
| $l_2$ | anticausal | anticausal | anticausal | anticausal | anticausal | anticausal | anticausal |
| $l_\infty$ | anticausal | anticausal | anticausal | anticausal | anticausal | anticausal | anticausal |

Table 1: Choice of norms for noise injection.

Figure 6: Effect for varying cluster numbers.

This process simulates the data generative process where $\boldsymbol{X}$ causes $Y$. Conversely, the instances in *synAnticausal* are generated similarly, but in the opposite direction, to reflect that $Y$ causes $\boldsymbol{X}$.

**Validate Disagreement Change for Anticausal and Causal Cases**  Fig. 4 demonstrates the change in disagreement with $10\%$, $20\%$ and $30\%$ label errors for synCausal and synAnticausal datasets. For the *synCausal* dataset, the disagreement remains unchanged with the increase of noise rates, and the regression coefficient $\hat{\beta}_1$ of the regression line is close to 0. This is because $Y'$ should be a random guess of noised $\tilde{Y}'$, which is proved in Theorem 1. On the other hand, for the *synAnticausal* dataset, there is a strong positive correlation between the disagreement and the noise level. In this case, $Y'$ is well estimated, and both $Y'$ and $\tilde{Y}$ are close to the latent (clean) class $Y$. When the noise level $\rho$ of our injected noise is increased to $0.5$, the modified label $\tilde{Y}^\rho$ becomes more seriously corrupted and tends to deviate far away from the observed label $\tilde{Y}$. This results in a larger disagreement between $\tilde{Y}^\rho$ and $Y'$. It is also observed that the regression coefficient becomes flatter when the label error is larger (e.g., Ins-30% and Sym-30%). Under this circumstance, a large number of original observable labels $\tilde{Y}$ are not identical to the latent clean class $Y$. As a result, $\tilde{Y}$ will be closer to a random guess of the clean class. Therefore the positive correlation between $\tilde{Y}$ and $Y'$ becomes weak. However, in these extreme settings, our estimator is still robust, because the regression coefficient of our regression line is still significantly different from $0$, and we can conclude that the dataset is anticausal.

**Robust to Label Errors**  Existing causal discovery methods primarily rely on checking the strength of dependence to infer causal relationships. Our method focuses on whether there is a change of dependence during the noise injection rather than the strength of the dependence. The strength of dependence can be easily influenced by label errors. As shown in the table in our appendix, with the increasing level of label errors, the performances of existing methods decrease. However, a consistent change of dependence can be observed (even if small) if $P(X)$ contains information about $P(\tilde{Y}|X)$. To further demonstrate this, we conducted additional experiments that let $P(X)$ and $P(\tilde{Y}|X)$ have very weak dependence on the anticausal dataset. This is achieved by manually adding high levels of label errors in $\tilde{Y}$ on the synthetic anticausal dataset. Specifically, we add 40% and 45% of label errors to synAnticausal dataset. Fig. 5 shows that, despite very small changes in dependence, with a regression coefficient (slope) of $0.075$, our method strongly rejects the null hypothesis that $P(X)$ does not contain information about $P(\tilde{Y}|X)$, with a p-value of 0.000.

**Influence of Different Norms for Noise Injection**  We investigate the performance of RoCA when using different norms to generate instance-dependent label noise for our noise injection on synthetic causal and anticausal datasets. The observed label contains different types (symmetric, instance) and

Table 2: Accuracy (%) for detecting the causal relation on different causal datasets.

| | GES | GIES | PC | ICD | RAI | FCI | LINGAM | CCDR | NOTEARS | NOTEARS-MLP | RoCA |
|---|---|---|---|---|---|---|---|---|---|---|---|
| | | | | Accuracy (%) on Anticausal Datasets | | | | | | | |
| *SynAnticausal* | **100.0** | **100.0** | **100.0** | **100.0** | 0.0 | **100.0** | 0.0 | 70.0 | 40.0 | **100.0** | **100.0** |
| *WDBC* | 20.0 | 20.0 | 50.0 | 90.0 | 0.0 | 40.0 | 80.0 | 0.0 | **100.0** | **100.0** | **100.0** |
| *Letter* | **100.0** | **100.0** | **100.0** | **100.0** | 0.0 | **100.0** | 100.0 | 40.0 | **100.0** | **100.0** | **100.0** |
| *Breastcancer* | 40.0 | 40.0 | 40.0 | 70.0 | 60.0 | 70.0 | 0.0 | 30.0 | 0.0 | 0.0 | **100.0** |
| *Coil* | 80.0 | 80.0 | 10.0 | 0.0 | 0.0 | 90.0 | 100.0 | 0.0 | 100.0 | 100.0 | **100.0** |
| *G241C* | 70.0 | 70.0 | 100.0 | 0.0 | 0.0 | 100.0 | 50.0 | 90.0 | 0.0 | 0.0 | **100.0** |
| *Iris* | 10.0 | 10.0 | 40.0 | 50.0 | 100.0 | 50.0 | 0.0 | 70.0 | 40.0 | **100.0** | **100.0** |
| *Mushroom* | **100.0** | **100.0** | 90.0 | 0.0 | 0.0 | 0.0 | 0.0 | 0.0 | 0.0 | 70.0 | 70.0 |
| *Segment* | 90.0 | 90.0 | 90.0 | **100.0** | 0.0 | **100.0** | 0.0 | 80.0 | 50.0 | **100.0** | **100.0** |
| *Usps* | 50.0 | 50.0 | 20.0 | 0.0 | 0.0 | 90.0 | 100.0 | 0.0 | 0.0 | 0.0 | **100.0** |
| *Waveform* | 20.0 | 20.0 | 100.0 | 100.0 | 0.0 | 100.0 | 100.0 | 20.0 | 0.0 | 10.0 | **100.0** |
| *Digit1* | 20.0 | 20.0 | 40.0 | 0.0 | 0.0 | 100.0 | 100.0 | 100.0 | 0.0 | 0.0 | **100.0** |
| *Pair0047* | 40.0 | 40.0 | NA | 40.0 | 0.0 | NA | 0.0 | 0.0 | 0.0 | 0.0 | **80.0** |
| Overall | 56.9 | 56.9 | 65.0 | 50 | 12.3 | 78.3 | 48.5 | 41.7 | 33.1 | 52.3 | **96.2** |

Table 3: Accuracy (%) for detecting the anticausal relation on anticausal datasets.

| | GES | GIES | PC | ICD | RAI | FCI | LINGAM | CCDR | NOTEARS | NOTEARS-MLP | RoCA |
|---|---|---|---|---|---|---|---|---|---|---|---|
| | | | | Accuracy (%) on Causal Datasets | | | | | | | |
| *SynCausal* | 70.0 | 70.0 | 40.0 | 0.0 | **100.0** | 0.0 | **100.0** | **100.0** | 70.0 | 70.0 | **100.0** |
| *Secstr* | 20.0 | 20.0 | 0.0 | 0.0 | 0.0 | 0.0 | **80.0** | **80.0** | 50.0 | 50.0 | 10.0 |
| *KrKp* | 0.0 | 0.0 | 30.0 | 10.0 | 0.0 | 0.0 | 0.0 | 40.0 | 50.0 | 70.0 | **100.0** |
| *Splice* | 0.0 | 0.0 | 0.0 | 0.0 | 0.0 | 0.0 | 0.0 | 0.0 | 40.0 | **50.0** | **50.0** |
| *Pair0070* | 0.0 | 0.0 | NA | 0.0 | 0.0 | NA | 0.0 | 0.0 | 0.0 | 0.0 | **50.0** |
| *Pair0071* | 0.0 | 0.0 | NA | 0.0 | 0.0 | NA | 0.0 | 0.0 | 0.0 | 0.0 | 0.0 |
| Overall | 15.0 | 15.0 | 17.5 | 1.7 | 16.7 | 0.0 | 30.6 | 36.7 | 35.0 | 40.0 | **51.6** |

Table 4: RoCA on large-scale noisy image datasets.

| *Clothing1M* | *CIFAR10* | *CIFAR10N* | | | | |
|---|---|---|---|---|---|---|
| | | Worst | Aggre | Random1 | Random2 | Random3 |
| $p = 0.0000$ **anticausal** | $p = 0.0000$ **anticausal** | $p = 0.0000$ **anticausal** | $p = 0.0000$ **anticausal** | $p = 0.0000$ **anticausal** | $p = 0.0000$ **anticausal** | $p = 0.0000$ **anticausal** |

levels of label error (10%, 20%, 30%). Fig. 1 shows that by changing the $l_1$ norm used in our paper to $l_0$, $l_2$, and $l_\infty$. For all settings, changes in norms do not affect RoCA's prediction of whether the dataset is causal or anticausal. The table including $p$-values is left in our appendix.

**Effect of Inconsistent Cluster Numbers**   We set the number of clusters equal to the number of different observed labels. Here, we show that making the number of clusters larger than the number of different observed labels results in unfaithful outcomes on SynCausal dataset. As shown in Fig. 6. The number of clusters and accuracy (1 - disagreements) are dependent. This contradicts the fact that $P(X)$ does not contain information about $P(\tilde{Y}|X)$. Intuitively, the reason is that in extreme cases, if the number of clusters equals the sample size, then the majority label within each cluster will be the observed label itself, and the accuracy will reach 100%.

## 4.2 PERFORMANCE OF ROCA ON REAL-WORLD DATASETS

We compare the RoCA method with other causal discovery algorithms in Table 2, Table 3 and Table 4. The accuracy (%) for detecting the anticausal relation on anticausal datasets is averaged over 10 different cases, including scenarios without label errors and those with different types of label errors (Instance 10%, Instance 20%, Instance 30%; Pair 10%, Pair 20%, Pair 30%; Sym 10%, Sym 20%, Sym 30%). We mark it as NA where the baseline cannot be employed to detect causal relations with only two variables. Notably, (1) the RoCA method is uniquely capable of being applied to large-scale image datasets *CIFAR10*, *CIFAR10N* and *Clothing1M*, which contain label errors, for the detection of causal and anticausal relations. The results demonstrate that our method is both accurate and robust. (2) For the *Pair0071* dataset, almost all methods misclassify it as an anticausal dataset. We believe this is due to the presence of a latent common cause affecting both the feature and the label. As acknowledged in their paper (Mooij et al., 2016), this scenario is possible.

## 5 CONCLUSION

This paper presents a scalable and robust estimator based on noise injection for determining causal and anticausal relations. The intuition is to leverage an information asymmetry between the distributions $P(\boldsymbol{X})$ and $P(\tilde{Y}|\boldsymbol{X})$ in anticausal and causal datasets. A practical estimator is proposed to verify this asymmetric property. Our theoretical analyses and empirical results demonstrate the effectiveness of the RoCA estimator in determining whether a dataset follows a causal or anticausal relationship.

ACKNOWLEDGMENT

BH was supported by RGC Young Collaborative Research Grant No. C2005-24Y and RIKEN Collaborative Research Fund. MG was supported by ARC DE210101624 and ARC DP240102088.

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

# Appendix

## Table of Contents

## A   A HYPOTHESIS TEST FOR THE REGRESSION COEFFICIENT

To rigorously validate the change of the disagreement, rather than directly evaluating if the regression coefficient $\hat{\beta}_1$ contained from Eq. (2) is near 0, we perform the one-sample $t$-test to quantify whether the regression coefficient $\hat{\beta}_1$ is significantly different from zero. We repeat the entire procedure of sampling different noise levels, calculating disagreements, and performing linear regressions 30 times to obtain a set of regression coefficient values. These regression coefficient values are then utilized in our hypothesis test to verify if the average regression coefficient is significantly different from 0.

Let $t^*$ be the observed value of the test statistic, $P_0$ denotes the t-distribution of the test statistic under the null hypothesis that the regression coefficient $\hat{\beta}_1$ is zero. Then the $p$-value of the $t$-test where $\hat{\beta}_1$ is significantly different from zero are as follows.

$$p = \mathbb{P}\left(T \geq t^* \mid T \sim P_0\right), \;\; t^* = \frac{\hat{\beta}_1 - 0}{\sqrt{\frac{1}{n-2} \frac{\sum_{i=1}^n \left(\ell_{01_i} - (\hat{\beta}_1 \rho_i + \hat{\beta}_0)\right)^2}{\sum_{i=1}^n (\rho_i - \bar{\rho})^2}} / \sqrt{n}}.$$

We check whether the $p$-value is less than the significance level $0.05$. If the condition holds, the null hypothesis will be rejected, indicating that the regression coefficient $\hat{\beta}_1$ is significantly different from zero, and the dataset is anticausal. Otherwise, the null hypothesis cannot be rejected, suggesting the regression coefficient $\hat{\beta}_1$ is zero. Then the dataset is very likely to be causal dataset.

## B   A REVIEW OF CAUSAL DISCOVERY METHODS

**Constraint-Based and Score-Based Approaches.**     To build a graph that captures these conditional independencies, the majority of constraint-based techniques look for conditional independencies in the empirical joint distribution. Since numerous graphs frequently satisfy a given set of conditional

dependencies, as was discussed above, constraint-based methods frequently produce a graph that represents some Markov equivalence classes. Unfortunately, large sample sizes are necessary for conditional independence tests to be reliable, and (Shah & Peters, 2020) highlights further difficulties in controlling Type I errors.

Score-based approaches test the validity of a candidate graph $\mathcal{G}$ according to some scoring function $S$. The goal is therefore stated as (Peters et al., 2017a):

$$\hat{\mathcal{G}} = \text{argmax}_{\mathcal{G} \text{ over } \mathbf{X}} S(\mathcal{D}, \mathcal{G}) \tag{4}$$

where the empirical data for the variables $\mathbf{X}$ is represented by $\mathcal{D}$. Common scoring functions include the Bayesian Information Criterion (BIC) (Geiger & Heckerman, 1994), the Minimum Description Length (as an approximation of Kolmogorov Complexity) (Janzing & Schölkopf, 2010; Grünwald & Vitányi, 2008; Kalainathan et al., 2020), the Bayesian Gaussian equivalent (BGe) score (Geiger & Heckerman, 1994), the Bayesian Dirichlet equivalence (BDe) score (Heckerman et al., 1995), the Bayesian Dirichlet equivalence uniform (BDeu) score (Heckerman et al., 1995), and others (Imoto et al., 2002; Hyvärinen & Smith, 2013; Huang et al., 2018).

**Functional Causal Models.** Methods based on causal function provide an alternate strategy for estimating causal effects. Assumptions about the data generation process are used in these causal function-based techniques. The causal function-based approach fits the causal function model among variables and then infers causal directions using causal assumptions, such as a non-Gaussian assumption of the noise (Shimizu et al., 2006; 2011) the independence assumption between cause variables and noise (Zhang & Hyvarinen, 2009; Peters et al., 2011; 2014) and the independence assumption between the distribution of cause variables and the causal function (Janzing et al., 2012). Most LiNGAM-based approaches for the linear case (Shimizu et al., 2006) assume non-Gaussian noise and linear causal relations between variables. This model seeks to determine a causal order among the random observed variables.

To deal with linear latent confounders, an estimation method utilizing overcomplete ICA (Lewicki & Sejnowski, 2000) is suggested. However, overcomplete ICA algorithms usually suffer from local optimum and cannot be employed when the number of variables is large.

By evaluating the independence between the estimated exogenous variables and the residual, (Tashiro et al., 2014) identify latent confounders. They discover that variables from subsets that are not impacted by latent confounders are included, and they estimate causal orders one at a time. (Chen & Chan, 2013) investigate linear non-Gaussian acyclic models in the presence of latent Gaussian confounders (LiNGAM-GC), which assumes that the latent confounders are Gaussian distributed independently.

## C CAUSAL GRAPHS AND STRUCTURAL CAUSAL MODELS (SCM)

Directed acyclic graphs (DAGs) serve as a formalism for representing causal relationships. In these graphs, arrows point from the parent node (direct cause) to the child node (direct effect) (Pearl, 2000). Building upon this graphical representation, a structural causal model (SCM) can be constructed to capture the causal mechanisms that underlie the data distribution.

An SCM is composed of a set of variables interconnected by functions, representing the flow of information. This model elucidates the causal relationships among variables, offering a detailed insight into the data generation process. Consider a DAG $G = (V, E)$ defined over a set of variables $\{X_1, X_2, \cdots, X_d, Y\}$, with $P$ representing their joint distribution. Let $\boldsymbol{X}$ be the set $\{X_1, X_2, \cdots, X_d\}$. The notation $\boldsymbol{X}_{PA_i^G}$ refers to the direct causes of $X_i$, while $\boldsymbol{Y}_{PA^G}$ denotes the direct causes of $Y$. Disturbances or errors in the generative processes of $X_i$ and $Y$ are represented by $N_i$ and $N_y$, respectively. The SCM for a classification dataset can be expressed as:

$$X_i := f_i(\boldsymbol{X}_{PA_i^G}, N_i), \ \ i = 1, ..., d; \ Y := f_y(\boldsymbol{Y}_{PA^G}, N_y).$$

The causal factorization of the joint distribution is given by:

$$P(\boldsymbol{X}, Y) = P(Y|\boldsymbol{Y}_{PA^G}) \prod_i P(X_i|\boldsymbol{X}_{PA_i^G}). \tag{5}$$

It's worth noting that both $\boldsymbol{X}_{PA_i^G}$ and $\boldsymbol{Y}_{PA^G}$ are allowed to be empty sets.

# D  EXPERIMENTS

## D.1  INTRODUCTION OF REAL-WORLD CAUSAL DATASETS

1. *KrKp* dataset contains 3196 instances with 36 attributes. Each instance is a board description for the chess endgame, where the feature attributes describe the board and the label determines whether it is "win" or "nowin". It is considered a causal dataset since the board description causally influences whether white will win.

2. *Splice* dataset contains 3190 instances with 60 attributes, where attributes describe sequential DNA nucleotide positions and the label is the type of splice sites. It is considered a causal dataset since the DNA sequence causes the splice sites.

3. *SecStr* dataset contains 83680 instances with 15 attributes, where attributes describe the amino acid and the label is the corresponding secondary chemical structure. It is considered a causal dataset since the secondary structure is determined by its amino acid features.

4. *Pair0070* dataset contains 4499 instances. It is a bi-variate dataset, where the feature describes parameters and the label contains the corresponding answers.It is considered as a causal dataset since parameters determine the answers.

5. *Pair0071* dataset contains 120 instances. It is a bi-variate dataset, where the feature describes symptoms and the label contains the corresponding classification of diseases. It is considered as a causal dataset since symptoms determine the type of diseases.

## D.2  INTRODUCTION OF REAL-WORLD ANTICAUSAL DATASETS

1. *WDBC* dataset contains 569 instances with 32 attributes. It is an anticausal dataset, where the class causes some of the tumor features.

2. *Breastcancer* dataset contains 286 instances with 9 attributes. It is an anticausal dataset, where the class causes some of the tumor features.

3. *Coil* dataset contains 1500 instances with 241 attributes. It is considered an anticausal/confounded dataset because the six-state class and the features are confounded by the 24-state variable of all objects.

4. *G241C* dataset contains 1500 instances with 241 attributes. It is considered an anticausal dataset since the class determines the features.

5. *Iris* dataset contains 150 instances with 4 attributes. It is an anticausal dataset, where the size of the plant is an effect of the category.

6. *Mushroom* dataset contains 8124 instances with 22 attributes. It is an anticausal dataset, where the attributes of the mushroom and the class are confounded by the mushroom taxonomy.

7. *Segment* dataset contains 2310 instances with 19 attributes. It is an anticausal dataset, where the class causes the features of the image.

8. *Usps* dataset contains 1500 instances with 240 attributes. It is an anticausal dataset, where the class and the features are confounded by the 10-state variable of all digits.

9. *Waveform* dataset contains 5000 instances with 21 attributes attributes and 1 label. Each class is generated from a combination of 2 or 3 "base" waves. It is considered an anticausal dataset since the class of the wave causes its attributes.

10. *Pair0047* dataset contains 255 instances. It is a bi-variate dataset, where the feature describes the number of cars and the label contains the type of day. It is considered as a causal dataset since number of cars determine the type of day.

11. *CIFAR10* dataset contains 60000 $32 \times 32$ color images (attributes) in 10 classes (label), with 6000 images per class. It is considered an anticausal dataset since the images are collected according to the predefined 10 different labels.

12. *CIFAR10N* has the same number of instances and attributes as those of *CIFAR10* while there are 5 different types of human-annotated real-world noisy labels from Amazon Mechanical Turk.

13. *Clothing1M* contains 1M clothing images in 14 classes. It is a causal dataset with noisy labels since the image determines its class and the data is collected from several online shopping websites.

14. *Digit1* dataset contains 1500 instances with 241 attributes. It is considered an anticausal dataset because the positive or negative angle and the features are confounded by the variable of continuous angle.

### D.3 INTRODUCTION OF BASELINE CAUSAL DISCOVERY METHODS

The baseline causal discovery methods we employed are as follows.

1. GES (Chickering, 2002): The Greedy Equivalence Search algorithm is a score-based Bayesian approach that heuristically searches for a graph that minimizes a likelihood score on the given data.
2. GIES (Hauser & Bühlmann, 2012): The Greedy Interventional Equivalence Search algorithm is similar to GES, but it incorporates interventional data for inference.
3. PC (Spirtes et al., 2000b): The Peter-Clark algorithm is one of the renowned score-based methods for causal discovery. It efficiently employs conditional tests on variables and variable sets.
4. ICD (Rohekar et al., 2021): Iterative Causal Discovery recovers causal graphs in the presence of latent confounders and selection bias. ICD relies on the causal Markov and faithfulness assumptions and identifies the equivalence class of the underlying causal graph.
5. RAI (Yehezkel & Lerner, 2009): Recursive Autonomy Identification learns the structure by sequentially applying conditional independence tests, edge direction, and structure decomposition into autonomous sub-structures.
6. FCI (Spirtes et al., 2000a): Fast Causal Inference stands out among constraint-based methods for its ability to detect latent confounders.
7. LiNGAM (Shimizu et al., 2006): Linear Non-Gaussian Acyclic Model assumes that there are no hidden confounders and all of the error terms are non-gaussian and detects causal relationships from observed data accordingly.
8. CCDR (Aragam & Zhou, 2015): Concave Penalized Coordinate Descent with Reparametrization is a fast, score-based method for learning Bayesian networks, utilizing sparse regularization and block-cyclic coordinate descent.
9. NOTEARS (Zheng et al., 2018): NOTEARS is a continuous optimization-based method for learning Directed Acyclic Graphs (DAGs) that represent causal relationships among variables. It reformulates the problem as a smooth optimization task with a differentiable acyclicity constraint.
10. NOTEARS-MLP (Zheng et al., 2020): NOTEARS-MLP extends the linear framework of NOTEARS to handle more complex causal structures by relaxing the linearity assumption. It leverages neural networks, specifically multi-layer perceptrons (MLPs), to represent nonlinear interactions between variables.

### D.4 MORE EXPERIMENTS ON REAL-WORLD DATASETS

In Table 6 and 7, we present the results of causal discovery obtained using our RoCA estimator compared to other baseline methods. Our RoCA estimator outperforms the baseline methods in accurately identifying the causal relationships. Among the 14 of 16 datasets, our RoCA estimator correctly identified the causal relationship between $X$ and $Y$ in the majority of cases. This holds even when the datasets contained different types of label noise, such as instance-dependent, pair, and symmetric noise, with noise rates ranging from $0\%$ to $30\%$. On the other hand, the performance of the baseline methods was generally satisfactory for anticausal datasets but lacked accuracy when dealing with causal datasets. This is because a causal dataset requires no features in $X$ to cause $Y$, which presents a challenge for these baseline methods. They need to ensure that there is no edge from any vertex representing features in $X$ pointing to the vertex representing $Y$ when recovering the causal diagram. Although these baseline methods tend to perform well in general tasks, they may not be suitable for this particular task, leading to misclassification of datasets as anticausal.

Furthermore, the time complexity of some baseline methods hinders their application to datasets with a large number of features, such as image datasets or datasets with hundreds of features (e.g., *G241C*, *Coil*, etc.). Completing the algorithm within a reasonable time frame becomes challenging for these methods. In this case, we classify the results as unknown.

Table 5: Comparing with other baselines on synthetic and real-world datasets (cont.).

| | Method | Original | Instance | | | Pair | | | Sym | | |
|---|---|---|---|---|---|---|---|---|---|---|---|
| | | 0% | 10% | 20% | 30% | 10% | 20% | 30% | 10% | 20% | 30% |
| SynCausal (causal) | GES | **causal** | **causal** | **causal** | anticausal | **causal** | **causal** | anticausal | **causal** | **causal** | anticausal |
| | GIES | **causal** | **causal** | **causal** | anticausal | **causal** | **causal** | anticausal | **causal** | **causal** | anticausal |
| | PC | **causal** | **causal** | anticausal | anticausal | **causal** | anticausal | anticausal | **causal** | anticausal | anticausal |
| | ICD | anticausal | anticausal | anticausal | anticausal | anticausal | anticausal | anticausal | anticausal | anticausal | anticausal |
| | RAI | **causal** | **causal** | **causal** | **causal** | **causal** | **causal** | **causal** | **causal** | **causal** | **causal** |
| | FCI | anticausal | anticausal | anticausal | anticausal | anticausal | anticausal | anticausal | anticausal | anticausal | anticausal |
| | LINGAM | **causal** | **causal** | **causal** | **causal** | **causal** | **causal** | **causal** | **causal** | **causal** | **causal** |
| | CCDR | **causal** | **causal** | **causal** | **causal** | **causal** | **causal** | **causal** | **causal** | **causal** | **causal** |
| | NOTEARS | **causal** | **causal** | anticausal | **causal** | **causal** | **causal** | anticausal | **causal** | **causal** | anticausal |
| | NOTEARS-MLP | **causal** | **causal** | anticausal | **causal** | **causal** | **causal** | anticausal | **causal** | **causal** | anticausal |
| | Our method | p=0.7886 **causal** | p=0.3131 **causal** | p=0.4466 **causal** | p=0.6729 **causal** | p=0.1399 **causal** | p=0.8041 **causal** | p=0.9772 **causal** | p=0.1399 **causal** | p=0.8041 **causal** | p=0.9772 **causal** |
| Secstr (causal) | GES | anticausal | anticausal | **causal** | **causal** | anticausal | anticausal | anticausal | anticausal | anticausal | anticausal |
| | GIES | anticausal | anticausal | **causal** | **causal** | anticausal | anticausal | anticausal | anticausal | anticausal | anticausal |
| | PC | anticausal | anticausal | anticausal | anticausal | anticausal | anticausal | anticausal | anticausal | anticausal | anticausal |
| | ICD | anticausal | anticausal | anticausal | anticausal | anticausal | anticausal | anticausal | anticausal | anticausal | anticausal |
| | RAI | unknown | unknown | unknown | unknown | unknown | unknown | unknown | unknown | unknown | unknown |
| | FCI | anticausal | anticausal | anticausal | anticausal | anticausal | anticausal | anticausal | anticausal | anticausal | anticausal |
| | LINGAM | **causal** | anticausal | anticausal | **causal** | **causal** | **causal** | **causal** | **causal** | **causal** | **causal** |
| | CCDR | **causal** | **causal** | anticausal | anticausal | **causal** | **causal** | **causal** | **causal** | **causal** | **causal** |
| | NOTEARS | **causal** | **causal** | anticausal | **causal** | **causal** | anticausal | unknown | **causal** | anticausal | unknown |
| | NOTEARS-MLP | **causal** | **causal** | anticausal | **causal** | **causal** | anticausal | unknown | **causal** | anticausal | unknown |
| | Our method | p=0.0000 anticausal | p=0.0000 anticausal | p=0.0000 anticausal | p=0.1510 **causal** | p=0.0000 anticausal | p=0.0000 anticausal | p=0.0000 anticausal | p=0.0000 anticausal | p=0.0000 anticausal | p=0.0000 anticausal |
| KrKp (causal) | GES | anticausal | anticausal | anticausal | anticausal | anticausal | anticausal | anticausal | anticausal | anticausal | anticausal |
| | GIES | anticausal | anticausal | anticausal | anticausal | anticausal | anticausal | anticausal | anticausal | anticausal | anticausal |
| | PC | **causal** | anticausal | anticausal | anticausal | **causal** | anticausal | anticausal | **causal** | anticausal | anticausal |
| | ICD | **causal** | anticausal | anticausal | anticausal | anticausal | anticausal | anticausal | anticausal | anticausal | anticausal |
| | RAI | anticausal | anticausal | anticausal | anticausal | anticausal | anticausal | anticausal | anticausal | anticausal | anticausal |
| | FCI | anticausal | anticausal | anticausal | anticausal | anticausal | anticausal | anticausal | anticausal | anticausal | anticausal |
| | LINGAM | anticausal | anticausal | anticausal | anticausal | anticausal | anticausal | anticausal | anticausal | anticausal | anticausal |
| | CCDR | **causal** | **causal** | unknown | unknown | **causal** | unknown | unknown | **causal** | unknown | unknown |
| | NOTEARS | **causal** | **causal** | **causal** | anticausal | **causal** | anticausal | unknown | **causal** | anticausal | unknown |
| | NOTEARS-MLP | **causal** | **causal** | **causal** | unknown | **causal** | **causal** | unknown | **causal** | **causal** | unknown |
| | Our method | p=0.3263 **causal** | p=0.7719 **causal** | p=0.6757 **causal** | p=0.2009 **causal** | p=0.4315 **causal** | p=0.1548 **causal** | p=0.3520 **causal** | p=0.4315 **causal** | p=0.1564 **causal** | p=0.3504 **causal** |
| Splice (causal) | GES | anticausal | anticausal | anticausal | anticausal | anticausal | anticausal | anticausal | anticausal | anticausal | anticausal |
| | GIES | anticausal | anticausal | anticausal | anticausal | anticausal | anticausal | anticausal | anticausal | anticausal | anticausal |
| | PC | anticausal | anticausal | anticausal | anticausal | anticausal | anticausal | anticausal | anticausal | anticausal | anticausal |
| | ICD | anticausal | anticausal | anticausal | anticausal | anticausal | anticausal | anticausal | anticausal | anticausal | anticausal |
| | RAI | unknown | unknown | unknown | unknown | unknown | unknown | unknown | unknown | unknown | unknown |
| | FCI | anticausal | anticausal | anticausal | anticausal | anticausal | anticausal | anticausal | anticausal | anticausal | anticausal |
| | LINGAM | anticausal | anticausal | anticausal | anticausal | anticausal | anticausal | anticausal | anticausal | anticausal | anticausal |
| | CCDR | anticausal | anticausal | anticausal | anticausal | anticausal | anticausal | anticausal | anticausal | anticausal | anticausal |
| | NOTEARS | **causal** | **causal** | anticausal | anticausal | **causal** | anticausal | anticausal | **causal** | anticausal | anticausal |
| | NOTEARS-MLP | **causal** | **causal** | **causal** | anticausal | **causal** | anticausal | anticausal | **causal** | anticausal | anticausal |
| | Our method | p=0.0022 anticausal | p=0.7749 **causal** | p=0.7748 **causal** | p=0.2731 **causal** | p=0.0395 anticausal | p=0.8958 **causal** | p=0.0000 anticausal | p=0.0085 anticausal | p=0.1976 **causal** | p=0.0314 anticausal |

Table 6: Comparing with baselines on synthetic and real-world datasets (cont.).

| | Method | Original | Instance | | | Pair | | | Sym | | |
|---|---|---|---|---|---|---|---|---|---|---|---|
| | | 0% | 10% | 20% | 30% | 10% | 20% | 30% | 10% | 20% | 30% |
| Pair0070 (causal) | GES | unknown | unknown | unknown | unknown | unknown | unknown | unknown | unknown | unknown | unknown |
| | GIES | unknown | unknown | unknown | unknown | unknown | unknown | unknown | unknown | unknown | unknown |
| | ICD | unknown | unknown | unknown | unknown | unknown | unknown | unknown | unknown | unknown | unknown |
| | RAI | unknown | unknown | unknown | unknown | unknown | unknown | unknown | unknown | unknown | unknown |
| | CCDR | unknown | unknown | unknown | unknown | unknown | unknown | unknown | unknown | unknown | unknown |
| | NOTEARS | unknown | unknown | unknown | unknown | unknown | unknown | unknown | unknown | unknown | unknown |
| | NOTEARS-MLP | unknown | unknown | unknown | unknown | unknown | unknown | unknown | unknown | unknown | unknown |
| | Our method | p=0.0663 | p=0.9966 | p=0.8995 | p=0.1843 | p=0.1791 | p=0.0000 | p=0.0158 | p=0.1791 | p=0.0000 | p=0.0158 |
| | | anticausal | **causal** | **causal** | **causal** | **causal** | anticausal | anticausal | **causal** | anticausal | anticausal |
| Pair0071 (causal) | GES | anticausal | anticausal | anticausal | anticausal | anticausal | anticausal | anticausal | anticausal | anticausal | anticausal |
| | GIES | anticausal | anticausal | anticausal | anticausal | anticausal | anticausal | anticausal | anticausal | anticausal | anticausal |
| | ICD | anticausal | anticausal | anticausal | anticausal | anticausal | anticausal | unknown | anticausal | anticausal | unknown |
| | RAI | anticausal | anticausal | anticausal | anticausal | anticausal | anticausal | anticausal | anticausal | anticausal | anticausal |
| | LINGAM | unknown | unknown | unknown | unknown | unknown | unknown | unknown | unknown | unknown | unknown |
| | CCDR | anticausal | anticausal | anticausal | anticausal | anticausal | anticausal | anticausal | anticausal | anticausal | anticausal |
| | NOTEARS | anticausal | anticausal | unknown | unknown | unknown | unknown | unknown | unknown | unknown | unknown |
| | NOTEARS-MLP | anticausal | anticausal | anticausal | anticausal | anticausal | anticausal | anticausal | anticausal | anticausal | anticausal |
| | Our method | p=0.0007 | p=0.0000 | p=0.0000 | p=0.0001 | p=0.0197 | p=0.0000 | p=0.0000 | p=0.0000 | p=0.0000 | p=0.0000 |
| | | anticausal | anticausal | anticausal | anticausal | anticausal | anticausal | anticausal | anticausal | anticausal | anticausal |
| SynAnticausal (anticausal) | GES | **anticausal** | **anticausal** | **anticausal** | **anticausal** | **anticausal** | **anticausal** | **anticausal** | **anticausal** | **anticausal** | **anticausal** |
| | GIES | **anticausal** | **anticausal** | **anticausal** | **anticausal** | **anticausal** | **anticausal** | **anticausal** | **anticausal** | **anticausal** | **anticausal** |
| | PC | **anticausal** | **anticausal** | **anticausal** | **anticausal** | **anticausal** | **anticausal** | **anticausal** | **anticausal** | **anticausal** | **anticausal** |
| | ICD | **anticausal** | **anticausal** | **anticausal** | **anticausal** | **anticausal** | **anticausal** | **anticausal** | **anticausal** | **anticausal** | **anticausal** |
| | RAI | causal | causal | causal | causal | causal | causal | causal | causal | causal | causal |
| | FCI | **anticausal** | **anticausal** | **anticausal** | **anticausal** | **anticausal** | **anticausal** | **anticausal** | **anticausal** | **anticausal** | **anticausal** |
| | LINGAM | unknown | unknown | unknown | unknown | unknown | unknown | unknown | unknown | unknown | unknown |
| | CCDR | **anticausal** | **anticausal** | **anticausal** | causal | **anticausal** | **anticausal** | causal | **anticausal** | **anticausal** | causal |
| | NOTEARS | **anticausal** | **anticausal** | causal | causal | **anticausal** | causal | causal | **anticausal** | causal | causal |
| | NOTEARS-MLP | **anticausal** | **anticausal** | **anticausal** | **anticausal** | **anticausal** | **anticausal** | **anticausal** | **anticausal** | **anticausal** | **anticausal** |
| | Our method | p=0.0000 | p=0.0000 | p=0.0000 | p=0.0000 | p=0.0000 | p=0.0000 | p=0.0000 | p=0.0000 | p=0.0000 | p=0.0000 |
| | | **anticausal** | **anticausal** | **anticausal** | **anticausal** | **anticausal** | **anticausal** | **anticausal** | **anticausal** | **anticausal** | **anticausal** |
| WDBC (anticausal) | GES | **anticausal** | causal | causal | causal | causal | causal | causal | causal | **anticausal** | causal |
| | GIES | **anticausal** | causal | causal | causal | causal | causal | causal | causal | **anticausal** | causal |
| | PC | **anticausal** | **anticausal** | **anticausal** | unknown | **anticausal** | unknown | unknown | **anticausal** | unknown | unknown |
| | ICD | **anticausal** | **anticausal** | **anticausal** | unknown | **anticausal** | **anticausal** | **anticausal** | **anticausal** | **anticausal** | **anticausal** |
| | RAI | unknown | unknown | unknown | unknown | unknown | unknown | unknown | unknown | unknown | unknown |
| | FCI | **anticausal** | **anticausal** | unknown | unknown | **anticausal** | unknown | unknown | **anticausal** | unknown | unknown |
| | LINGAM | **anticausal** | unknown | unknown | unknown | unknown | unknown | unknown | unknown | unknown | unknown |
| | CCDR | causal | causal | causal | causal | causal | causal | causal | causal | causal | causal |
| | NOTEARS | **anticausal** | **anticausal** | **anticausal** | **anticausal** | **anticausal** | **anticausal** | **anticausal** | **anticausal** | **anticausal** | **anticausal** |
| | NOTEARS-MLP | **anticausal** | **anticausal** | **anticausal** | **anticausal** | **anticausal** | **anticausal** | **anticausal** | **anticausal** | **anticausal** | **anticausal** |
| | Our method | p=0.0000 | p=0.0000 | p=0.0000 | p=0.0000 | p=0.0000 | p=0.0000 | p=0.0000 | p=0.0000 | p=0.0000 | p=0.0000 |
| | | **anticausal** | **anticausal** | **anticausal** | **anticausal** | **anticausal** | **anticausal** | **anticausal** | **anticausal** | **anticausal** | **anticausal** |
| Letter (anticausal) | GES | **anticausal** | **anticausal** | **anticausal** | **anticausal** | **anticausal** | **anticausal** | **anticausal** | **anticausal** | **anticausal** | **anticausal** |
| | GIES | **anticausal** | **anticausal** | **anticausal** | **anticausal** | **anticausal** | **anticausal** | **anticausal** | **anticausal** | **anticausal** | **anticausal** |
| | PC | **anticausal** | **anticausal** | **anticausal** | **anticausal** | **anticausal** | **anticausal** | **anticausal** | **anticausal** | **anticausal** | **anticausal** |
| | ICD | **anticausal** | **anticausal** | **anticausal** | **anticausal** | **anticausal** | **anticausal** | **anticausal** | **anticausal** | **anticausal** | **anticausal** |
| | RAI | unknown | unknown | unknown | unknown | unknown | unknown | unknown | unknown | unknown | unknown |
| | FCI | **anticausal** | **anticausal** | **anticausal** | **anticausal** | **anticausal** | **anticausal** | **anticausal** | **anticausal** | **anticausal** | **anticausal** |
| | LINGAM | **anticausal** | **anticausal** | **anticausal** | **anticausal** | **anticausal** | **anticausal** | **anticausal** | **anticausal** | **anticausal** | **anticausal** |
| | CCDR | unkown | unkown | causal | causal | **anticausal** | **anticausal** | causal | **anticausal** | unknown | **anticausal** |
| | NOTEARS | **anticausal** | **anticausal** | **anticausal** | **anticausal** | **anticausal** | **anticausal** | **anticausal** | **anticausal** | **anticausal** | **anticausal** |
| | NOTEARS-MLP | **anticausal** | **anticausal** | **anticausal** | **anticausal** | **anticausal** | **anticausal** | **anticausal** | **anticausal** | **anticausal** | **anticausal** |
| | Our method | p=0.0000 | p=0.0000 | p=0.0000 | p=0.0000 | p=0.0000 | p=0.0000 | p=0.0000 | p=0.0000 | p=0.0000 | p=0.0000 |
| | | **anticausal** | **anticausal** | **anticausal** | **anticausal** | **anticausal** | **anticausal** | **anticausal** | **anticausal** | **anticausal** | **anticausal** |
| Breastcancer (anticausal) | GES | **anticausal** | unknown | **anticausal** | unknown | unknown | **anticausal** | unknown | unknown | **anticausal** | unknown |
| | GIES | **anticausal** | unknown | **anticausal** | unknown | unknown | **anticausal** | unknown | unknown | **anticausal** | unknown |
| | PC | **anticausal** | unknown | **anticausal** | unknown | **anticausal** | **anticausal** | unknown | unknown | **anticausal** | unknown |
| | ICD | **anticausal** | **anticausal** | **anticausal** | unknown | **anticausal** | **anticausal** | unknown | **anticausal** | **anticausal** | unknown |
| | RAI | **anticausal** | **anticausal** | **anticausal** | **anticausal** | **anticausal** | unknown | unknown | **anticausal** | unknown | unknown |
| | FCI | **anticausal** | **anticausal** | **anticausal** | unknown | **anticausal** | **anticausal** | unknown | **anticausal** | **anticausal** | unknown |
| | LINGAM | unknown | unknown | unknown | unknown | unknown | unknown | unknown | unknown | unknown | unknown |
| | CCDR | causal | causal | causal | **anticausal** | causal | causal | **anticausal** | causal | causal | **anticausal** |
| | NOTEARS | unknown | unknown | unknown | unknown | unknown | unknown | unknown | unknown | unknown | unknown |
| | NOTEARS-MLP | causal | causal | causal | unknown | causal | causal | unknown | causal | causal | unknown |
| | Our method | p=0.0000 | p=0.0000 | p=0.0000 | p=0.0000 | p=0.0000 | p=0.0000 | p=0.0000 | p=0.0000 | p=0.0000 | p=0.0000 |
| | | **anticausal** | **anticausal** | **anticausal** | **anticausal** | **anticausal** | **anticausal** | **anticausal** | **anticausal** | **anticausal** | **anticausal** |
| Coil (anticausal) | GES | **anticausal** | causal | **anticausal** | **anticausal** | **anticausal** | causal | **anticausal** | **anticausal** | **anticausal** | **anticausal** |
| | GIES | **anticausal** | causal | **anticausal** | **anticausal** | **anticausal** | causal | **anticausal** | **anticausal** | **anticausal** | **anticausal** |
| | PC | causal | unknown | causal | unknown | unknown | **anticausal** | causal | causal | unknown | unknown |
| | ICD | unknown | unknown | unknown | unknown | unknown | unknown | unknown | unknown | unknown | unknown |
| | RAI | unknown | unknown | unknown | unknown | unknown | unknown | unknown | unknown | unknown | unknown |
| | FCI | **anticausal** | **anticausal** | **anticausal** | unknown | **anticausal** | **anticausal** | **anticausal** | **anticausal** | **anticausal** | **anticausal** |
| | LINGAM | **anticausal** | **anticausal** | **anticausal** | **anticausal** | **anticausal** | **anticausal** | **anticausal** | **anticausal** | **anticausal** | **anticausal** |
| | CCDR | causal | causal | unknown | unknown | causal | causal | causal | causal | causal | causal |
| | NOTEARS | **anticausal** | **anticausal** | **anticausal** | **anticausal** | **anticausal** | **anticausal** | **anticausal** | **anticausal** | **anticausal** | **anticausal** |
| | NOTEARS-MLP | **anticausal** | **anticausal** | **anticausal** | **anticausal** | **anticausal** | **anticausal** | **anticausal** | **anticausal** | **anticausal** | **anticausal** |
| | Our method | p=0.0000 | p=0.0000 | p=0.0000 | p=0.0000 | p=0.0000 | p=0.0000 | p=0.0000 | p=0.0000 | p=0.0000 | p=0.0000 |
| | | **anticausal** | **anticausal** | **anticausal** | **anticausal** | **anticausal** | **anticausal** | **anticausal** | **anticausal** | **anticausal** | **anticausal** |
| G241C (anticausal) | GES | **anticausal** | **anticausal** | **anticausal** | **anticausal** | **anticausal** | **anticausal** | **anticausal** | causal | causal | causal |
| | GIES | **anticausal** | **anticausal** | **anticausal** | **anticausal** | **anticausal** | **anticausal** | **anticausal** | causal | causal | causal |
| | PC | **anticausal** | **anticausal** | unknown | **anticausal** | **anticausal** | **anticausal** | **anticausal** | **anticausal** | **anticausal** | **anticausal** |
| | ICD | unknown | unknown | unknown | unknown | unknown | unknown | unknown | unknown | unknown | unknown |
| | RAI | unknown | unknown | unknown | unknown | unknown | unknown | unknown | unknown | unknown | unknown |
| | FCI | **anticausal** | **anticausal** | **anticausal** | **anticausal** | **anticausal** | **anticausal** | **anticausal** | **anticausal** | **anticausal** | **anticausal** |
| | LINGAM | **anticausal** | unknown | causal | unknown | **anticausal** | causal | **anticausal** | unknown | **anticausal** | **anticausal** |
| | CCDR | **anticausal** | **anticausal** | **anticausal** | **anticausal** | **anticausal** | **anticausal** | **anticausal** | **anticausal** | causal | **anticausal** |
| | NOTEARS | unknown | unknown | unknown | unknown | unknown | unknown | unknown | unknown | unknown | unknown |
| | NOTEARS-MLP | unknown | unknown | unknown | unknown | unknown | unknown | unknown | unknown | unknown | unknown |
| | Our method | p=0.0000 | p=0.0000 | p=0.0000 | p=0.0000 | p=0.0000 | p=0.0000 | p=0.0000 | p=0.0000 | p=0.0000 | p=0.0000 |
| | | **anticausal** | **anticausal** | **anticausal** | **anticausal** | **anticausal** | **anticausal** | **anticausal** | **anticausal** | **anticausal** | **anticausal** |

Table 7: Comparing with baselines on synthetic and real-world datasets.

| Dataset | Method | Original | Instance | | | Pair | | | Sym | | |
|---|---|---|---|---|---|---|---|---|---|---|---|
| | | 0% | 10% | 20% | 30% | 10% | 20% | 30% | 10% | 20% | 30% |
| Iris (anticausal) | GES | anticausal | unknown | unknown | unknown | unknown | unknown | unknown | unknown | unknown | unknown |
| | GIES | anticausal | unknown | unknown | unknown | unknown | unknown | unknown | unknown | unknown | unknown |
| | PC | anticausal | unknown | anticausal | unknown | anticausal | unknown | unknown | anticausal | causal | unknown |
| | ICD | anticausal | unknown | anticausal | unknown | anticausal | unknown | unknown | anticausal | anticausal | unknown |
| | RAI | anticausal | anticausal | anticausal | anticausal | anticausal | anticausal | anticausal | anticausal | anticausal | anticausal |
| | FCI | anticausal | unknown | anticausal | unknown | anticausal | unknown | unknown | anticausal | anticausal | unknown |
| | LINGAM | unknown | unknown | unknown | unknown | unknown | unknown | unknown | unknown | unknown | unknown |
| | CCDR | anticausal | anticausal | anticausal | causal | anticausal | causal | causal | anticausal | anticausal | anticausal |
| | NOTEARS | anticausal | unknown | unknown | unknown | anticausal | anticausal | unknown | anticausal | unknown | unknown |
| | NOTEARS-MLP | anticausal | anticausal | anticausal | anticausal | anticausal | anticausal | anticausal | anticausal | anticausal | anticausal |
| | Our method | p=0.0000 | p=0.0000 | p=0.0000 | p=0.0000 | p=0.0000 | p=0.0000 | p=0.0000 | p=0.0000 | p=0.0000 | p=0.0000 |
| | | anticausal | anticausal | anticausal | anticausal | anticausal | anticausal | anticausal | anticausal | anticausal | anticausal |
| Mushroom | GES | anticausal | anticausal | anticausal | anticausal | anticausal | anticausal | anticausal | anticausal | anticausal | anticausal |
| | GIES | anticausal | anticausal | anticausal | anticausal | anticausal | anticausal | anticausal | anticausal | anticausal | anticausal |
| | PC | causal | anticausal | anticausal | anticausal | anticausal | anticausal | anticausal | anticausal | anticausal | anticausal |
| | ICD | unknown | unknown | unknown | unknown | unknown | unknown | unknown | unknown | unknown | unknown |
| | RAI | unknown | unknown | unknown | unknown | unknown | unknown | unknown | unknown | unknown | unknown |
| | LINGAM | unknown | unknown | unknown | unknown | unknown | unknown | unknown | unknown | unknown | unknown |
| | CCDR | unknown | unknown | unknown | unknown | unknown | unknown | unknown | unknown | unknown | unknown |
| | NOTEARS | causal | unknown | unknown | unknown | unknown | unknown | unknown | unknown | unknown | unknown |
| | NOTEARS-MLP | anticausal | anticausal | anticausal | unknown | anticausal | anticausal | unknown | anticausal | anticausal | unknown |
| | Our method | p=0.0000 | p=0.0625 | p=0.0000 | p=0.0000 | p=0.0000 | p=0.0000 | p=0.0865 | p=0.0000 | p=0.0000 | p=0.0865 |
| | | anticausal | causal | anticausal | anticausal | anticausal | anticausal | causal | anticausal | anticausal | causal |
| Segment (anticausal) | GES | unknown | anticausal | anticausal | anticausal | anticausal | anticausal | anticausal | anticausal | anticausal | anticausal |
| | GIES | unknown | anticausal | anticausal | anticausal | anticausal | anticausal | anticausal | anticausal | anticausal | anticausal |
| | PC | unknown | anticausal | anticausal | anticausal | anticausal | anticausal | anticausal | anticausal | anticausal | anticausal |
| | ICD | anticausal | anticausal | anticausal | anticausal | anticausal | anticausal | anticausal | anticausal | anticausal | anticausal |
| | RAI | unknown | unknown | unknown | unknown | unknown | unknown | unknown | unknown | unknown | unknown |
| | FCI | anticausal | anticausal | anticausal | anticausal | anticausal | anticausal | anticausal | anticausal | anticausal | anticausal |
| | LINGAM | unknown | unknown | unknown | unknown | unknown | unknown | unknown | unknown | unknown | unknown |
| | CCDR | causal | causal | anticausal | anticausal | anticausal | anticausal | anticausal | anticausal | anticausal | anticausal |
| | NOTEARS | anticausal | anticausal | causal | causal | causal | causal | causal | anticausal | anticausal | anticausal |
| | NOTEARS-MLP | anticausal | anticausal | anticausal | anticausal | anticausal | anticausal | anticausal | anticausal | anticausal | anticausal |
| | Our method | p=0.0000 | p=0.0000 | p=0.0000 | p=0.0000 | p=0.0000 | p=0.0000 | p=0.0000 | p=0.0000 | p=0.0000 | p=0.0000 |
| | | anticausal | anticausal | anticausal | anticausal | anticausal | anticausal | anticausal | anticausal | anticausal | anticausal |
| Usps (anticausal) | GES | anticausal | anticausal | causal | anticausal | causal | anticausal | causal | causal | anticausal | causal |
| | GIES | anticausal | anticausal | causal | anticausal | causal | anticausal | causal | causal | anticausal | causal |
| | PC | causal | unknown | unknown | unknown | causal | anticausal | unknown | causal | anticausal | unknown |
| | ICD | unknown | unknown | unknown | unknown | unknown | unknown | unknown | unknown | unknown | unknown |
| | RAI | unknown | unknown | unknown | unknown | unknown | unknown | unknown | unknown | unknown | unknown |
| | FCI | anticausal | anticausal | anticausal | unknown | anticausal | anticausal | anticausal | anticausal | anticausal | anticausal |
| | LINGAM | anticausal | anticausal | anticausal | anticausal | anticausal | anticausal | anticausal | anticausal | anticausal | anticausal |
| | CCDR | causal | causal | causal | causal | causal | causal | causal | causal | causal | causal |
| | NOTEARS | unknown | unknown | unknown | unknown | unknown | unknown | unknown | unknown | unknown | unknown |
| | NOTEARS-MLP | unknown | unknown | unknown | unknown | unknown | unknown | unknown | unknown | unknown | unknown |
| | Our method | p=0.0000 | p=0.0000 | p=0.0000 | p=0.0000 | p=0.0000 | p=0.0000 | p=0.0000 | p=0.0000 | p=0.0000 | p=0.0000 |
| | | anticausal | anticausal | anticausal | anticausal | anticausal | anticausal | anticausal | anticausal | anticausal | anticausal |
| Waveform (anticausal) | GES | anticausal | causal | causal | causal | causal | causal | causal | causal | anticausal | causal |
| | GIES | anticausal | causal | causal | causal | causal | causal | causal | causal | anticausal | causal |
| | PC | anticausal | anticausal | anticausal | anticausal | anticausal | anticausal | anticausal | anticausal | anticausal | anticausal |
| | ICD | anticausal | anticausal | anticausal | anticausal | anticausal | anticausal | anticausal | anticausal | anticausal | anticausal |
| | RAI | unknown | unknown | unknown | unknown | unknown | unknown | unknown | unknown | unknown | unknown |
| | FCI | anticausal | anticausal | anticausal | anticausal | anticausal | anticausal | anticausal | anticausal | anticausal | anticausal |
| | LINGAM | anticausal | anticausal | anticausal | anticausal | anticausal | anticausal | anticausal | anticausal | anticausal | anticausal |
| | CCDR | causal | causal | causal | causal | causal | anticausal | causal | causal | anticausal | causal |
| | NOTEARS | unknown | unknown | unknown | unknown | unknown | unknown | unknown | unknown | unknown | unknown |
| | NOTEARS-MLP | anticausal | unknown | unknown | unknown | unknown | unknown | unknown | unknown | unknown | unknown |
| | Our method | p=0.0000 | p=0.0000 | p=0.0000 | p=0.0000 | p=0.0000 | p=0.0000 | p=0.0000 | p=0.0000 | p=0.0000 | p=0.0000 |
| | | anticausal | anticausal | anticausal | anticausal | anticausal | anticausal | anticausal | anticausal | anticausal | anticausal |
| Digit1 (anticausal) | GES | anticausal | causal | causal | anticausal | causal | causal | causal | causal | causal | causal |
| | GIES | anticausal | causal | causal | anticausal | causal | causal | causal | causal | causal | causal |
| | PC | causal | anticausal | anticausal | causal | anticausal | unknown | unknown | anticausal | unknown | unknown |
| | ICD | unknown | unknown | unknown | unknown | unknown | unknown | unknown | unknown | unknown | unknown |
| | RAI | unknown | unknown | unknown | unknown | unknown | unknown | unknown | unknown | unknown | unknown |
| | FCI | anticausal | anticausal | anticausal | anticausal | anticausal | anticausal | anticausal | anticausal | anticausal | anticausal |
| | LINGAM | anticausal | anticausal | anticausal | anticausal | anticausal | anticausal | anticausal | anticausal | anticausal | anticausal |
| | CCDR | anticausal | anticausal | anticausal | anticausal | anticausal | anticausal | anticausal | anticausal | anticausal | anticausal |
| | NOTEARS | unknown | unknown | unknown | unknown | unknown | unknown | unknown | unknown | unknown | unknown |
| | NOTEARS-MLP | unknown | unknown | unknown | unknown | unknown | unknown | unknown | unknown | unknown | unknown |
| | Our method | p=0.0000 | p=0.0000 | p=0.0000 | p=0.0000 | p=0.0000 | p=0.0000 | p=0.0000 | p=0.0000 | p=0.0000 | p=0.0000 |
| | | anticausal | anticausal | anticausal | anticausal | anticausal | anticausal | anticausal | anticausal | anticausal | anticausal |
| Pair0047 (anticausal) | GES | anticausal | anticausal | anticausal | anticausal | unknown | unknown | unknown | unknown | unknown | unknown |
| | GIES | anticausal | anticausal | anticausal | anticausal | unknown | unknown | unknown | unknown | unknown | unknown |
| | ICD | anticausal | anticausal | anticausal | anticausal | unknown | unknown | unknown | unknown | unknown | unknown |
| | RAI | unknown | unknown | unknown | unknown | unknown | unknown | unknown | unknown | unknown | unknown |
| | LINGAM | unknown | unknown | unknown | unknown | unknown | unknown | unknown | unknown | unknown | unknown |
| | CCDR | causal | causal | causal | causal | causal | causal | causal | causal | causal | causal |
| | NOTEARS | unknown | unknown | unknown | unknown | unknown | unknown | unknown | unknown | unknown | unknown |
| | NOTEARS-MLP | unknown | unknown | unknown | unknown | unknown | unknown | unknown | unknown | unknown | unknown |
| | Our method | p=0.0000 | p=0.0000 | p=0.0000 | p=0.0000 | p=0.0000 | p=0.0000 | p=0.0000 | p=0.0000 | p=0.0000 | p=0.0000 |
| | | anticausal | anticausal | anticausal | anticausal | anticausal | anticausal | anticausal | anticausal | anticausal | anticausal |

## D.5 INFLUENCE OF NORM CHOICE TO ROCA

| | | | | SynCausal | | | |
|---|---|---|---|---|---|---|---|
| | 0% | Ins-10% | Ins-20% | Ins-30% | Sym-10% | Sym-20% | Sym-30% |
| $l_0$ | causal ($p = 0.109$) | causal ($p = 0.100$) | causal ($p = 0.106$) | causal ($p = 0.105$) | causal ($p = 0.688$) | causal ($p = 0.607$) | causal ($p = 0.135$) |
| $l_2$ | causal ($p = 0.198$) | causal ($p = 0.414$) | causal ($p = 0.136$) | causal ($p = 0.277$) | causal ($p = 0.965$) | causal ($p = 0.874$) | causal ($p = 0.111$) |
| $l_\infty$ | causal ($p = 0.399$) | causal ($p = 0.258$) | causal ($p = 0.154$) | causal ($p = 0.104$) | causal ($p = 0.901$) | causal ($p = 0.946$) | causal ($p = 0.164$) |

| | | | | SynAnticausal | | | |
|---|---|---|---|---|---|---|---|
| | 0% | Ins-10% | Ins-20% | Ins-30% | Sym-10% | Sym-20% | Sym-30% |
| $l_0$ | anticausal ($p = 0.000$) | anticausal ($p = 0.000$) | anticausal ($p = 0.000$) | anticausal ($p = 0.000$) | anticausal ($p = 0.000$) | anticausal ($p = 0.000$) | anticausal ($p = 0.000$) |
| $l_2$ | anticausal ($p = 0.000$) | anticausal ($p = 0.000$) | anticausal ($p = 0.000$) | anticausal ($p = 0.000$) | anticausal ($p = 0.000$) | anticausal ($p = 0.000$) | anticausal ($p = 0.000$) |
| $l_\infty$ | anticausal ($p = 0.000$) | anticausal ($p = 0.000$) | anticausal ($p = 0.000$) | anticausal ($p = 0.000$) | anticausal ($p = 0.000$) | anticausal ($p = 0.000$) | anticausal ($p = 0.000$) |

Table 8: Performance of RoCA when using different norms to generate instance-dependent label noise for our noise injection on synthetic causal and anticausal datasets.

## E PSEDUOCODE OF OUR INSTANCE-DEPENDENT NOISE GENERATION

---
**Algorithm 1** Generation of Instance-dependent Noisy Labels
---
**Require:** An average noise level $\rho$; A sample $S = \{(X_i, \tilde{Y}_i)\}_{i=0}^m$, where contains $C$ number of classes, and $X \in \mathbb{R}^d$.
1: Initialize an empty list $A$ with length $m$.
2: **for** each $i^{th}$ example $(x_i, \tilde{y}) \in S$: **do**
3:     Let $a_i = ||X_i||_1$ and add $a_i$ into $A$.
4: **end for**
5: Sort the values in $A$ in ascending order.
6: Sample a vector $P \in \mathbb{R}^m$ from a $m$-dimensional truncated normal distribution with mean $\rho$, upper limit 1, and lower limit 0.
7: Sort the values in $P$ in ascending order.
8: **for** $i$ in range $(0, m)$: **do**
9:     Let the individual flip rate of the $i^{th}$ example $\rho_{x_i} =$ (the $i^{th}$ element in $P$).
10: **end for**
11: Generate the instance-dependent noisy label of the $i^{th}$ example $\tilde{Y}^{\rho_{x_i}}$ using the flip rate $\rho_{x_i}$.
---

## F PROOFS

In this section, we show all the proofs. We remind some notations first.

- Let $\mathcal{X}$ be the instance space and $C$ the set of all possible classes.

- Let $S = \{(x_i, \tilde{y}_i)\}_{t=0}^m$ be an sample set.

- Let $h : \mathcal{X} \rightarrow \{1, \ldots, C\}$, be a hypothesis that predicts pseudo labels of instances. Concretely, it can be a K-means algorithm together with the Hungarian algorithm which matches the cluster ID to the corresponding pseudo labels. Let $\mathcal{H}$ be the hypothesis space, where $h \in \mathcal{H}$.

- Let $\tilde{R}^\rho(h) = \mathbb{E}_{(\boldsymbol{x}, \tilde{y}^\rho) \sim P(\boldsymbol{X}, \tilde{Y}^\rho)}[\mathbb{1}_{\{h(\boldsymbol{x}) \neq \tilde{y}^\rho\}}]$ be the expected disagreement $\tilde{R}(h)$ between pseudo labels and generated labels $\tilde{y}^\rho$ with $\rho$-level noise injection.

- Let $\hat{\tilde{R}}_S^\rho(h)$ be the average disagreement (or empirical risk) of $h$ on the set $S$ after $\rho$-level noise injection.

Firstly, we illustrate the Rademacher complexity bound.

**Definition 3** (The Rademacher Complexity Bound (Mohri et al., 2018)). Let $\mathcal{H}$ be a family of functions taking values in $\{-1, +1\}$, and let $\mathcal{D}$ be the distribution over the input space $\mathcal{X}$. Then, for any $\delta > 0$, with probability at least $1 - \delta/2$ over a sample $S = (x_1, \ldots, x_m)$ of size $m$ drawn

according to $\mathcal{D}$, for any function $h \in \mathcal{H}$,

$$\hat{R}_S(h) - R(h) \leq 2\hat{\mathfrak{R}}_S(\mathcal{H}) + 3\sqrt{\frac{\log\frac{4}{\delta}}{2n}}, \tag{6}$$

where $R(h)$ is the expected risk of the function $h$, and $\hat{R}_S(h)$ is the empirical risk of the function $h$ on the sample $S$ (Mohri et al., 2018). Specifically, let $c$ be a target concept, then,

$$R(h) = \mathbb{E}_{x \sim \mathcal{D}}[\mathbb{1}_{\{h(x_i) \neq c(x_i)\}}], \ \hat{R}_S(h) = \frac{1}{m}\sum_{i=1}^{m}\mathbb{1}_{\{h(x_i) \neq c(x_i)\}}.$$

### F.1 PROOF OF THEOREM 1

*Proof.* Under causal setting, $h$ random guess the clean labels, i.e., $\forall i, j \in C \wedge i \neq j \wedge \forall t \in \{0, 1, \ldots, m\}$, $P(Y' = y'|Y = y, X = x) = \frac{1}{C}$. Then we will prove that if $h$ can only random guess the clean labels, then $h$ can only random guess the observed labels that contain the label error, i.e., $\tilde{R}(h, x) = \frac{C-1}{C}$.

By the assumption that for every instance and clean class pair $(x, y)$, its observed label $\tilde{y}$ is obtained by a noise rate $\rho_x$ such that $P(\tilde{Y} = \tilde{y}|Y = y, \boldsymbol{X} = x) = \frac{\rho_x}{C-1}$ for all $\tilde{y} \neq y \wedge \tilde{y} \in C$, the risk $\tilde{R}(h, x)$ of $h$ on $x$ and its observed labels comes from two parts:

- When $h$ misclassifies the clean label, $h$ also misclassifies the observed label, i.e., ($y \neq y'$ and $y' \neq \tilde{y}$).

- When $h$ successfully classifies the clean label, $h$ misclassifies the observed label, i.e., ($y = y'$ and $y' \neq \tilde{y}$).

Specifically, the expected risk of each example is as follows.

$$\begin{aligned}
\tilde{R}(h, x) &= R(h, x)(1 - \frac{\rho_x}{C-1}) + (1 - R(h, x))\rho_x \\
&= \frac{C-1}{C}\frac{C-1-\rho_x}{C-1} + \frac{\rho_x}{C} \\
&= \frac{C-1-\rho_x}{C} + \frac{\rho_x}{C} \\
&= \frac{C-1}{C}.
\end{aligned} \tag{7}$$

Because our noise is designed to also satisfy the assumption, after injecting our designed instance-dependent noise, the risk $\hat{\tilde{R}}_S^{\rho^1}(h)$ and $\hat{\tilde{R}}_S^{\rho^2}(h)$ under two different (expected) noise levels $\rho^1 = \mathbb{E}_x[\rho_x^1]$ and $\rho^2 = \mathbb{E}_x[\rho_x^2]$ does not change under the anticausal setting.

$$\tilde{R}^{\rho^1}(h, x) = \tilde{R}(h, x)(1 - \frac{\rho_x^1}{C-1}) + (1 - \tilde{R}(h, x))\rho_x^1 = \frac{C-1}{C}. \tag{8}$$

$$\tilde{R}^{\rho^2}(h, x) = \tilde{R}(h, x)(1 - \frac{\rho_x^2}{C-1}) + (1 - \tilde{R}(h, x))\rho_x^2 = \frac{C-1}{C}. \tag{9}$$

The above equations show that after injecting two different levels of instance-dependent label noise, the risks do not change. For completeness, we also illustrate the convergence rate of the difference between two empirical risks with respect to sample size. By employing the Rademacher complexity

bound, with a probability $1 - \delta/2$,

$$\hat{\tilde{R}}_S^{\rho^1}(h) \leq \mathbb{E}_x[\tilde{R}^{\rho^1}(h, x)] + 2\hat{\mathfrak{R}}_S(\mathcal{H}) + 3\sqrt{\frac{\log\frac{4}{\delta}}{2m}}$$

$$= \tilde{R}^{\rho^1}(h) + 2\hat{\mathfrak{R}}_S(\mathcal{H}) + 3\sqrt{\frac{\log\frac{4}{\delta}}{2m}},$$

similarly,

$$\hat{\tilde{R}}_S^{\rho^2}(h) \leq \mathbb{E}_x[\tilde{R}^{\rho^2}(h, x)] + 2\hat{\mathfrak{R}}_S(\mathcal{H}) + 3\sqrt{\frac{\log\frac{4}{\delta}}{2m}}$$

$$= \tilde{R}(h, \rho^2) + 2\hat{\mathfrak{R}}_S(\mathcal{H}) + 3\sqrt{\frac{\log\frac{4}{\delta}}{2m}}.$$

By applying the symmetric property of the Rademacher complexity bound to the above two inequalities, with a probability $1 - 2\delta$,

$$|\hat{\tilde{R}}_S^{\rho^1}(h) - \tilde{R}^{\rho^1}(h)| \leq 2\hat{\mathfrak{R}}_S(\mathcal{H}) + 3\sqrt{\frac{\log\frac{4}{\delta}}{2m}} \text{ and}$$

$$|\hat{\tilde{R}}_S^{\rho^2}(h) - \tilde{R}(h, \rho^2)| \leq 2\hat{\mathfrak{R}}_S(\mathcal{H}) + 3\sqrt{\frac{\log\frac{4}{\delta}}{2m}}.$$

Combining the above two inequalities, we get

$$|\hat{\tilde{R}}_S^{\rho^1}(h) - \tilde{R}^{\rho^1}(h) - \hat{\tilde{R}}_S^{\rho^2}(h) + \tilde{R}(h, \rho^2)| \leq 4\hat{\mathfrak{R}}_S(\mathcal{H}) + 6\sqrt{\frac{\log\frac{4}{\delta}}{2m}}.$$

By Eq. 8 and Eq. 9, the expected risk $\tilde{R}^{\rho^1}(h) = \mathbb{E}_X\left[\tilde{R}^{\rho^1}(h, x)\right] = \mathbb{E}_X\left[\frac{C-1}{C}\right]$ and $\tilde{R}(h, \rho^2) = \mathbb{E}_X\left[\tilde{R}^{\rho^2}(h, x)\right] = \mathbb{E}_X\left[\frac{C-1}{C}\right]$ both equals to $\frac{C-1}{C}$, then the above inequality becomes

$$|\hat{\tilde{R}}_S^{\rho^1}(h) - \hat{\tilde{R}}_S^{\rho^2}(h)| \leq 4\hat{\mathfrak{R}}_S(\mathcal{H}) + 6\sqrt{\frac{\log\frac{4}{\delta}}{2m}}, \tag{10}$$

with a probability $1\text{-}2\delta$, which completes the proof. $\qquad\square$

### F.2 PROOF OF THEOREM 2

*Proof.* The expected risk on the observed label for each instance $x$ is that:

$$\tilde{R}^\rho(h, x) = \tilde{R}(h, x)(\frac{\rho_x}{C-1}) + (1 - \tilde{R}(h, x))(1 - \rho_x)$$

$$= \rho_x + \tilde{R}(h, x) - \rho_x\tilde{R}(h, x) - \frac{\rho_x\tilde{R}(h, x)}{C-1}.$$

Then the expected risk of the distribution of observed data is that:

$$\tilde{R}^\rho(h) = \mathbb{E}_X\left[\tilde{R}^\rho(h,x)\right]$$

$$= \mathbb{E}_X\left[\rho_x + \tilde{R}(h,x) - \rho_x\tilde{R}(h,x) - \frac{\rho_x\tilde{R}(h,x)}{C-1}\right]$$

$$= \mathbb{E}_X\left[\tilde{R}(h,x)\right] + \mathbb{E}_X\left[\rho_x - \rho_x\tilde{R}(h,x) - \frac{\rho_x\tilde{R}(h,x)}{C-1}\right]$$

$$= \tilde{R}(h) + \mathbb{E}_X\left[\left(1 - \tilde{R}(h,x) - \frac{\tilde{R}(h,x)}{C-1}\right)\rho_x\right]$$

$$= \tilde{R}(h) + \mathbb{E}_X\left[\left(1 - \frac{C\tilde{R}(h,x)}{C-1}\right)\rho_x\right]. \tag{11}$$

Moving $\tilde{R}(h)$ to the LHS of the above equation completes the proof. □

Note that the convergence rate of $\mathbb{E}\left[\left(1 - \frac{C\tilde{R}(h,x)}{C-1}\right)\rho_x\right]$ can also be directly derived by replacing the expected risks with the empirical risks in Eq. 11. This process employs Inequality 6 and shares a similar conceptual foundation as the proof for the coverage rate presented in Theorem 1.

## G    DISCUSSION ON ASSUMPTIONS

Our method is based on some common assumptions in causal discovery: causal minimality, absence of latent confounders, and independent causal mechanisms (Peters et al., 2014). To ensure that the disagreements under different noise levels remain constant in a causal setting when employing RoCA, we need an additional assumption to constrain the types of label errors in datasets. Specifically, it assumes that for every instance and clean class pair $(x, y)$, the observed label $\tilde{y}$ is derived with a noise rate $\rho_x$ such that $P(\tilde{Y} = \tilde{y}|Y = y, \boldsymbol{X} = x) = \frac{\rho_x}{C-1}$ for all $\tilde{y} \neq y \wedge \tilde{y} \in C$. Note that most types of label errors defined in previous work satisfy this conditional, including random classification label errors (Wang et al., 2019), asymmetric label errors (Yao et al., 2020), manifold label errors (Cheng et al., 2022), and part-dependent label errors (Xia et al., 2020). Note that, pairflip label errors (Han et al., 2018) and instance-dependent label errors (Yuan et al., 2023; Lin et al., 2024; Wang et al., 2024) do not satisfy this condition. In this case, our method cannot be directly applied. To utilize our method, existing techniques for learning with label errors can be applied first to estimate clean labels. Specifically, methods such as those proposed by Liu & Tao (2016) and Patrini et al. (2017) can be used initially. These methods are statistically consistent, ensuring that the clean labels can be uniquely identified. Once we have the estimated clean labels, RoCA can then be applied.

The effectiveness of pseudo-label generation through clustering depends on several assumptions and the suitability of the chosen clustering method for the given data. In our work, we employ two clustering methods: K-means and a self-supervised clustering method SPICE* (Niu et al., 2021), each with specific requirements. K-means clustering relies on the Low-Density Separation assumption that cluster boundaries pass through areas of low probability density in input space for guaranteed classification performance (Pollard, 1981). For image data, achieving low-density separation in the raw input space can be challenging due to the high-dimensional and complex nature of such data. To address the limitations of K-means for image data, we leverage a self-supervised clustering method. Existing theoretical studies (Saunshi et al., 2019; HaoChen et al., 2021; Von Kügelgen et al., 2021) show that if semantically similar points are sampled from the same latent class, the representation learned by minimizing a self-supervised loss function can guarantee classification performance. Specifically, these works define semantically similar points mathematically from different perspectives such as distributional distance (Saunshi et al., 2019) and constraints for data augmentation (HaoChen et al., 2021; Von Kügelgen et al., 2021; Hong et al., 2024; Zheng et al., 2024), with analyses conducted under both finite and infinite examples. Computational costs of our method are primarily raised by multiple steps of pseudo-label generation.

