# OpenReview forum: "A Robust Method to Discover Causal or Anticausal Relation"
_ICLR.cc/2025/Conference — ICLR 2025 Poster_

### Official Review · Reviewer_nDtS · 2024-10-30

**Soundness:** 3
**Presentation:** 4
**Contribution:** 3
**Rating:** 6
**Confidence:** 3

**Summary:**

This paper proposes a robust estimator, the Robust Causal and Anticausal (RoCA) Estimator, to identify causal or anticausal relationships in datasets, even when label noise is present. Traditional methods for causal discovery struggle with high-dimensional data, especially when label noise disrupts the feature-label relationships. The RoCA estimator's key innovation is a noise injection technique that exploits an asymmetry between causal and anticausal relationships. In causal datasets, instance distribution P(X) should not help predict the label distribution P(Y∣X), while in anticausal datasets, P(X) can be informative for predicting P(Y∣X). Key to this method is the identification of an asymmetric property: in causal datasets, the instance distribution P(X) lacks predictive information about the noisy label distribution P(Y~∣X), while in anticausal datasets, P(X) does provide such information. By injecting varying levels of instance-dependent label noise and measuring shifts in the disagreement rate between pseudo-labels (generated through clustering) and noisy labels, RoCA detects causality based on whether disagreement trends change across noise levels. Unlike traditional methods that can struggle with perceptual data or noise, RoCA remains accurate across various noise types and dataset scales, as validated in experiments on both synthetic and real-world datasets.

**Strengths:**

The paper introduces the RoCA (Robust Causal and Anticausal) Estimator, an innovative approach to causality detection that is both noise-robust and adaptable to complex, high-dimensional data. This originality stems from its focus on the asymmetric relationship between instance distributions P(X) and noisy label distributions P(Y~∣X), rather than relying on cleaner, idealized data. By targeting causality detection under instance-dependent label noise, the paper introduces a novel technique to address causal or anti-causal discovery in reql-world noisy data. This subimission os overall high-quality. RoCA’s robustness across multiple noise types is carefully validated, including tests on both synthetic and real-world datasets. Moreover, the paper's detailed experiments showcase significant improvement over baseline methods, lending credibility to RoCA’s real-world applicability. Furthermore, the proposed method is backed-up by sound theoretical analyisis, showing that RoCA can indeed detect causal and anti-causal relations. The paper is structured and articulated in a clear and accessible manner. Technical details about clustering, disagreement measures, and the rationale behind examining instance distribution asymmetries are explained systematically. Overall, this work is significant for the field of Causal Inference. RoCA is potentially applicable to fields requiring robust causal inference from noisy data, such as healthcare, economics, and social sciences, where data noise is often unavoidable.

**Weaknesses:**

The robustness of RoCA relies heavily on certain parameters, such as those governing the clustering process and the threshold for disagreement trends. If these parameters are not tuned appropriately, the method’s performance may degrade, particularly in cases of subtle causal effects or in datasets with less distinct noise patterns. This reliance on careful parameter tuning may limit the method’s effectiveness in some real-world applications where optimal parameter settings are difficult to identify. Furthermore, RoCA’s reliance on clustering as a step in handling noisy labels presupposes that the dataset is large enough to form meaningful clusters. In cases of limited or highly sparse data, this dependency may affect the method’s reliability, as insufficient data could lead to unstable or non-representative clusters, impacting the overall accuracy of causality detection.

**Questions:**

Regarding the weaknesses mentioned above, can you comment on the robustness of RoCA to parameters tuning? Also, what is the performance of RoCA in low-sample regimes?

---

> ### Author Response · Authors · 2024-11-25
> **Response from Authors**
>
> **Q1. Regarding the weaknesses mentioned above, can you comment on the robustness of RoCA to parameter tuning?**
>
> A1. We follow your constructive comment to conduct additional experiments on the SynAnticausal and SynCausal datasets by varying the hyperparameters of the backbone clustering methods.
> Specifically, we changed the used cluster centroid selection strategies from 'k-means++' (which selects initial centroids based on an empirical probability distribution of the points' contributions to the overall inertia) to 'random' (which chooses $n\_clusters$ observations at random from the data). We also adjusted the maximum iterations to 5, 10, 50, and 100. We injected L1-based instance-dependent noise into the datasets. The sample size is 20,000.
>
> | Dataset | Noise Type | Noise Level | Max Iterations | Result |
> |---------------|--------------------|-------------|----------------|------------|
> | SynAnticausal | Instance-Dependent | 0% | 5 | Anticausal |
> | SynAnticausal | Instance-Dependent | 10% | 10 | Anticausal |
> | SynAnticausal | Symmetric Flipping | 20% | 50 | Anticausal |
> | SynCausal | Instance-Dependent | 0% | 5 | Causal |
> | SynCausal | Instance-Dependent | 10% | 10 | Causal |
> | SynCausal | Symmetric Flipping | 20% | 50 | Causal |
>
> The results demonstrate that our method remains effective in determining causal and anticausal relationships under these varying conditions. The reason is that the condition for our method to fail is that on an anticausal dataset, if the regression coefficient of the disagreement line is very close to 0, which means that the cluster method is a completely random guess (under different injected levels), then $P(X)$ contains no information about $P(Y|X)$.
>
> ---
>
> **Q2. What is the performance of RoCA in low-sample regimes?**
>
> A2. We have conducted additional experiments with only 50, 100, and 200 examples on the SynAnticausal and SynCausal datasets. The table below summarizes how predictions vary under different noise types and sample sizes:
>
> |                    | IDN-10%  | IDN-20% | IDN-30% | Sym-10% | Sym-20% | Sym-30% |
> |--------------------|----------|---------|---------|---------|---------|---------|
> | **SynAnticausal (50)**  | Anticausal   | Causal  | Causal  | Causal  | Causal  | Causal  |
> | **SynAnticausal (100)** | Anticausal | Anticausal | Anticausal | Anticausal | Causal | Causal |
> | **SynAnticausal (200)** | Anticausal | Anticausal | Anticausal | Anticausal | Causal | Causal |
> | **SynCausal (50)**      | Causal   | Causal  | Causal  | Causal  | Causal  | Causal  |
> | **SynCausal (100)**     | Causal   | Causal  | Causal  | Causal  | Causal  | Causal  |
> | **SynCausal (200)**     | Causal   | Causal  | Causal  | Causal  | Causal  | Causal  |
>
>
>
> The results demonstrate that when the sample size is too small, the variance of the regression coefficients from disagreement lines, obtained through multiple trials, increases significantly. This results in a low p-value, making it difficult to confidently determine that the clustering method is not merely guessing randomly.

---

> > ### Comment · Reviewer_nDtS · 2024-11-26
> >
> > Thank you for your reply. After reading the rebuttal, my score remains unchanged.

---

> > > ### Author Response · Authors · 2024-11-30
> > > **Thanks**
> > >
> > > Dear Reviewer nDtS,
> > >
> > > Thank you for your positive support and help improve our paper. We believe that the experiments on parameter tuning and low-sample scenarios are very meaningful, and we will include them in the final version of the paper.
> > >
> > > Best regards,
> > > The Authors

---

### Official Review · Reviewer_pE7x · 2024-11-01

**Soundness:** 2
**Presentation:** 2
**Contribution:** 3
**Rating:** 6
**Confidence:** 3

**Summary:**

This paper aims to discern if a data generation process is causal or anti-causal. It proposed the Robust Causal and Anticausal (RoCA) Estimator, where the authors attempt to differentiate the two by investigating if the instance distribution, $P(X)$, offers information about the prediction task, $P(Y|X)$. They opted for the noisy class-posterior distribution, $P(\tilde{Y}|X)$, to surrogate for $P(Y|X)$ and used clustering to generate pseudo labels. Their discussion suggests that there should be no correlation between mismatch and noise levels in a causal scenario, while in an anti-causal context, a correlation should exist. Experiments are conducted on several causal discovery datasets and several image datasets.

**Strengths:**

1. The problem of interest is an important topic in casual discovery.

2. The idea of using injected label noise to discover causal vs. anti causal relationships is novel.

3. Experiment results show significant improvement over existing causal discovery approaches.

**Weaknesses:**

1. The proposed algorithm crucially depends on the idea that $P(\tilde{Y}|X)$ is a surrogate of $P(Y|X)$. Unfortunately, it is not clear what information the surrogate must preserve for the $P(Y|X)$. An intuitive algorithm for constructing pseudo label and injecting label noise is used in the implementation, but I think the paper lacks a formal discussion of the identifiability. The authors can provide a precise definition on under what conditions RoCA can identify the direction.

2. In the instance-dependent noise injection, the "dependence" between the instance and its noisy label is created based on the l1-norm. Because instances with different $X$ can have the same l1-norm, the noisy label only depends on the l1-norm but not the original instance. For the noise to be instance dependent, the noisy label should be related to the semantic meaning of the instance, e.g., a husky dog is incorrectly labelled as a wolf, or a lion cub is incorrectly labelled as a cat. The authors should make a better explanation of how the noise-generation process relates to the noise being instance-dependent.

3. It seems that the validity of the algorithm will largely depend on $P(X)$ as a crucial step in the reasoning is that "under the causal setting, $P(X)$ does not contain information about the surrogate distribution $P(\tilde{Y}|X)$." and thus $P(X)$ cannot help predict observed labels. But, if we consider a dataset where there are 10 classes, and the samples of these 10 classes happen to reside in well-defined clusters. In other words, a clustering algorithm (e.g., k-means) can easily separate these 10 clusters. Then, the pseudo label obtained from $P(X)$ will not be random guesses of the observed labels. Even if noise is introduced to the observed labels, the pseudo label could continue to correspond to the observed labels unless the noise rate is extremely high.

This does not suggest an algorithm should work on all scenarios. But rather, as a causal discovery algorithm, the authors should be clear regarding when the algorithm is expected to work, and when the algorithm is expected to not work/fail. Currently, this is not clearly discussed in the paper.

**Questions:**

Besides the weaknesses, there are also some questions regarding the clarity of writing.

1. For the expected noise level injection using the truncated distribution. What is the specific procedure of truncation? If you set the mean to 0.2 and truncate at 0 by setting all negative values to 0, then the actual expected noise level will be smaller than 0.2. How is this accounted for in the experiments?

2. What is the chosen unsupervised algorithm for clustering the instances to generate pseudo labels? Also, if the clustering algorithm changes, then the cluster results will also change. How will this affect RoCA?

3. In the real-world noisy image datasets, how to validate whether the results are correct? CIFAR-10N is a dataset injected with human-annotated label errors. As the human will annotate the image by looking at the image, the data generation process seems to be causal $X\to \tilde{Y}$ instead of anticausal. For Clothing1M, the dataset contains images crawled from the web, and the labels are generated from the text description accompanying the image, it also seems to be causal. How do we determine whether a real-world noisy dataset is causal or anti-causal for the sake of evaluating performance?

---

> ### Author Response · Authors · 2024-11-25
> **Response from Authors**
>
> **Q1. The authors should make a better explanation of how the noise-generation process relates to the noise being instance-dependent.**
>
> A1. Thank you very much for such an insightful comment. We realize that we should emphasize this to avoid confusion, we have added the following paragraph in our revised version.
>
> For the UCI datasets, the $l_1$-norm is directly applied to the covariates, which represent meaningful features of each instance. For image datasets, where raw pixel values do not capture meaningful semantic features, we employ a Variational Autoencoder (VAE) to extract feature representations that encapsulate the semantic meaning of each instance. The $l_1$-norm noise is then applied to these feature representations rather than the raw pixel values. In this setup, the latent dimension of the VAE is set to 10, and a ResNet-18 is used as the backbone for robust and accurate feature extraction.
>
> ---
>
> **Q2. It seems that the validity of the algorithm will largely depend on  P(X)  as a crucial step in the reasoning is that "under the causal setting, P(X) does not contain information about the surrogate distribution P(\tilde{Y}|X)." and thus P(X) cannot help predict observed labels. But, if we consider a dataset where there are 10 classes, and the samples of these 10 classes happen to reside in well-defined clusters. In other words, a clustering algorithm (e.g., k-means) can easily separate these 10 clusters. Then, the pseudo label obtained from P(X)  will not be random guesses of the observed labels. Even if noise is introduced to the observed labels, the pseudo label could continue to correspond to the observed labels unless the noise rate is extremely high.**
>
> A2. (assumption clarification) The claim that "under the causal setting, $P(X)$ does not contain information about the surrogate distribution $P(\tilde{Y}|X)$" is directly derived from the fundamental **Independent Causal Mechanisms (ICM) principle** [1] and is not an assumption proposed in our paper. We appreciate your question and recognize the importance of emphasizing this point to avoid confusion. In the revised version of the paper, we will explicitly highlight this point.
>
> (example clarification) In the example you constructed, the assumption that the cluster assumption holds directly implies that $P(X)$ can be leveraged to help learn the labels. This indicates that $P(X)$ contains information about $P(Y|X)$. Such a dependency suggests that the dataset is **anticausal**, where the causal direction is $Y \to X$, or there exists a common cause influencing both $X$ and $Y$.
>
> (Further explanation) Here we would like to provide a further explanation for the assumption and example completeness.
>
> - "under the causal setting, P(X) does not contain information about the surrogate distribution P(\tilde{Y}|X)." it aligns directly with the Independent Causal Mechanisms (ICM) principle. The ICM [1] states that  “The causal generative process (...) is composed of independent and autonomous modules that do not inform or influence each other.”
>
> - As explained in the paragraph Independent Causal Mechanisms of Section 2.2 in [2]
> In probabilistic terms, this means that the conditional distributions of each variable given its causal parents, P(Xi∣PAi)P(X_i | \text{PA}_i), are independent and do not share any information. However, this independence is violated in non-causal or anticausal settings, where the mechanisms are not independent  (see also an example in Figure 1).
> - In the example you constructed, by assuming clusters hold, the independence between $P(X)$ and $P(Y|X)$ is directly violated, as $P(X)$ contains information about $P(Y|X)$. This dependency implies that the causal direction is not $X \to Y$, but rather $Y \to X$ or that $X$ and $Y$ share a common cause. Therefore, the validity of leveraging $P(X)$ for predicting labels relies on the dataset being anticausal or non-causal, and the example provided is consistent with this anticausal interpretation.

---

> ### Author Response · Authors · 2024-11-25
> **Response from Authors**
>
> **Q3. Unfortunately, it is not clear what information the surrogate must preserve for the P(Y|X). An intuitive algorithm for constructing pseudo label and injecting label noise is used in the implementation, but I think the paper lacks a formal discussion of the identifiability. The authors can provide a precise definition on under what conditions RoCA can identify the direction.**
>
> A3. We believe that this question is directly related to your Q2. In A2 we have explained that  "under the causal setting, $P(X)$ does not contain information about the surrogate distribution $P(\tilde{Y}|X)$" is directly derived from the fundamental **Independent Causal Mechanisms (ICM) principle**
>
>
> There we would like to follow your constructive comments and add a clear statement of assumptions for the clustering method can guarantee classification performance in the revised version. Specifically
>
> The effectiveness of pseudo-label generation through clustering depends on several assumptions and the suitability of the chosen clustering method for the given data. In our work, we employ two clustering methods: **K-means** and a self-supervised clustering method **SPICE**, each with specific requirements and theoretical support.
> Conventional K-means clustering relies on the Low-Density Separation assumption that class or cluster boundaries pass through areas of low probability density in input space for guaranteed classification performance [2].
> For image data, achieving low-density separation in the raw input space can be challenging due to the high-dimensional and complex nature of such data. To address the limitations of K-means for image data, we leverage a self-supervised clustering method.
> Existing theoretical studies [2, 3, 4] show that if semantically similar points are sampled from the same latent class, the representation learned by minimizing a self-supervised loss function can guarantee classification performance. Specifically, these works define semantically similar points mathematically from different perspectives such as distributional distance [2] and constraints for data augmentation [3, 4], with analyses conducted under both finite and infinite examples.
> Thanks for raising this point which improves our paper.
>
>
> ---
>
> **Q4.  What is the chosen unsupervised algorithm for clustering the instances to generate pseudo labels? Also, if the clustering algorithm changes, then the cluster results will also change. How will this affect RoCA?**
>
> A4. We followed your constructive comment and conducted extensive experiments on the synCausal and synAnticausal datasets, each containing 2,000 examples and incorporating different types of label errors—Symmetry Flipping, Pair Flipping, and Instance-Dependent Label Errors—at error levels of 10%, 20%, and 30%.
>
> Specifically, we evaluated the method using various clustering algorithms:
> - **Agglomerative Clustering (AC):** Recursively merges pairs of clusters in the sample data using linkage distance.
> - **Gaussian Mixture Models (GMM):** A soft clustering machine learning method used to determine the probability that each data point belongs to a given cluster.
> - **Spectral Clustering (SC):** A graph-based technique that utilizes spectral decomposition to reveal the structural properties of a graph by capturing similarities among data points.
> In addition, we experimented with different norms for noise injection, specifically $L_1$, $L_2$, and $L_\infty$ norms.
>
> Across all these settings, our method consistently and successfully identified causal and anticausal relationships using a significance level of $p = 0.05$. The stability of the method did not diminish when alternative unsupervised clustering algorithms were employed for pseudo-label generation. This robustness suggests that the method is not dependent on a specific clustering algorithm like K-means or SPICE*, but can generalize well across different clustering techniques.
>
> The following tables summarize our findings for both datasets.
>
>
> ### **Table 1: Results on synCausal Dataset**
>
> | **Error Types (Including Three Levels)** | **AC Accuracy** | **GMM Accuracy** | **SC Accuracy** |
> |------------------------------------------|-----------------|------------------|-----------------|
> | Sym | 100% | 100% | 100% |
> | Pair | 100% | 100% | 100% |
> | Ins | 100% | 100% | 100% |
> ---
>
> ### **Table 2: Results on synAnticausal Dataset**
>
> | **Error Types (Including Three Levels)** | **AC Accuracy** | **GMM Accuracy** | **SC Accuracy** |
> |------------------------------------------|-----------------|------------------|-----------------|
> | Sym | 100% | 100% | 100% |
> | Pair | 100% | 100% | 100% |
> | Ins | 100% | 100% | 100% |

---

> ### Author Response · Authors · 2024-11-25
> **Response from Authors**
>
> **Q5.  For the expected noise level injection using the truncated distribution. What is the specific procedure of truncation? If you set the mean to 0.2 and truncate at 0 by setting all negative values to 0, then the actual expected noise level will be smaller than 0.2. How is this accounted for in the experiments?**
>
> A5. Thank you for your time and the detailed review! It would have no influence on our results, as we do not use the mean value of the truncated distribution. Instead, we recalculate the actual noise level directly from the data for our calculations.
>
> *We have added your nice example to the revised version of our paper as follows.*
>
> Note that the mean value of a truncated Gaussian distribution does not reflect the actual noise level. For example, if its mean value is set to 0.2 and truncated at 0, the actual expected noise level will be smaller than 0.2. Therefore, in our calculation of the regression coefficient for disagreement under different noise levels, we use the actual noise level computed directly from the data rather than relying on the mean value of the truncated distribution.
> Thanks for helping improve our paper.
>
> ---
>
> **Q6. In the real-world noisy image datasets, how to validate whether the results are correct? CIFAR-10N is a dataset injected with human-annotated label errors. As the human will annotate the image by looking at the image, the data generation process seems to be causal X \to \tilde{Y}  instead of anticausal. For Clothing1M, the dataset contains images crawled from the web, and the labels are generated from the text description accompanying the image, it also seems to be causal. How do we determine whether a real-world noisy dataset is causal or anti-causal for the sake of evaluating performance?**
>
> A6. We would like to kindly clarify that our aim is to infer the causal relationship between the **ground truth labels** $Y$ and the input data $X$ by making use of noisy observed labels $\tilde{Y}$, rather than inferring the causal relationship between $\tilde{Y}$ and $X$.
>
> To determine whether a real-world noisy dataset is causal or anticausal for evaluating performance, we consider the principle of **Independent Causal Mechanisms (ICM)** [1]: when we follow the causal factorization of the joint distribution, intervening (changing) in one distribution does not affect other distributions.
>
> In the case of image classification datasets like CIFAR-10N, the class label $Y$ (e.g., "cat", "dog") causally influence the features of the image $X$. Changing the class directly affects the shape, appearance, and other characteristics captured in the image. This means that the distribution of image features $P(X)$ depends on the class label $Y$, implying that $Y \rightarrow X$. Therefore, the dataset is considered **anticausal**.
>
> Note that some image datasets could be considered causal. For instance, in medical imaging, the physical characteristics of a tumor seen in MRI scans (size, shape, texture) directly cause the classification of the tumor type (benign or malignant). In this case, $X \rightarrow Y$ because changing the classification criteria (defined by humans) does not influence the physical appearance of the tumor.
>
> ---
>
> ### References
>
> [1]. Heinze-Deml, Christina, and Nicolai Meinshausen. "Conditional variance penalties and domain shift robustness." Machine Learning 110.2 (2021): 303-348.
>
> [2]. Kügelgen, Julius, et al. "Semi-supervised learning, causality, and the conditional cluster assumption." UAI'20.

---

> > ### Author Response · Authors · 2024-11-25
> >
> > Dear Reviewer pE7x,
> >
> > Thank you for having taken your time to provide us with your valuable comments, which help improves our papr. This is a gentle reminder that the discussion period is nearing its conclusion, we hope you have taken the time to consider our responses to your review. If you have any additional questions or concerns, please let us know so we can resolve them before the discussion period concludes. If you feel our responses have satisfactorily addressed your concerns, it would be greatly appreciated if you could raise your score to show that the existing concerns have been addressed.
> >
> > Thank you!
> >
> > The Authors

---

> > > ### Comment · Reviewer_pE7x · 2024-11-26
> > >
> > > Thanks for the detailed responses. Most of my comments are well addressed and thus I am happy to adjust my rating to 6.
> > >
> > > On Q1 I have some further concerns regarding using vanilla VAE to construct instance dependent noises. For a vanilla VAE, there is actually no guarantee that the features learned are semantic meaningful. Even in the empirical exploration of beta-VAE and variants, most of the disentanglement result requires training encoders from scratch, using a pre trained backbone may be counterproductive for learning meaningful representations in VAE manner as their targets are fundamentally different; even then, Locatello et al theoretically showed that disentanglement is impossible without additional information. I think these can be considered to make the paper more sound.

---

> ### Author Response · Authors · 2024-11-30
> **Thanks for the positive support**
>
> Dear Reviewer pE7x,
>
> We apologize for sending our response during the weekend. Thank you for your thoughtful feedback and for adjusting your rating. We greatly appreciate your detailed observations on Q1. We acknowledge that using disentangled representations could be a better solution, as they are not only instance-dependent but also semantically dependent.  Therefore,
>
> - We have highlighted this point in the appendix, stating:  *"Using iVAE could further enhance the label noise dependence on semantic features."*
>
> - We have conducted additional experiments using iVAE as a backbone on three image datasets with varying noise rates. The experiment results are included in the appendix. The results are consistent and confirm the detection of these datasets as anticausal. Since all results point to anticausal datasets, we have omitted the table in this response.
>
>
> Thank you again for your valuable input and help improve our paper.
>
> Best regards,
>
> The Authors

---

### Official Review · Reviewer_gqJM · 2024-11-03

**Soundness:** 3
**Presentation:** 4
**Contribution:** 3
**Rating:** 8
**Confidence:** 4

**Summary:**

To address the limitations of existing methods on high-dimensional data (such as images), this paper proposes a novel noise injection method that determines the causal direction by examining the asymmetry in data distribution. The proposed RoCA estimator uses unsupervised clustering to generate pseudo-labels and determines the causal direction of the dataset by observing the divergence between pseudo-labels and observed labels under varying degrees of noise injection. This approach performs well in both theoretical analysis and empirical experiments, effectively distinguishing causal and anti-causal data generation processes across multiple datasets.

**Strengths:**

1. The method proposed in this paper differs from traditional causal discovery algorithms, providing a new direction for high-dimensional causal relationship detection.

2. The effectiveness of the RoCA method is demonstrated through theoretical analysis and validation on multiple datasets.

3. The appendix provides a detailed review of relevant techniques and datasets in the field, offering strong reference value.

4. By generating pseudo-labels through unsupervised clustering, RoCA can effectively determine causal direction without relying on true labels, making this method more adaptable and flexible for various application scenarios.

**Weaknesses:**

1.	How can we ensure that the pseudo-labels generated by clustering effectively represent the true distribution of the data? The paper lacks analysis or at least a clear statement of assumptions in this regard, which would strengthen the theoretical support for the method’s pseudo-label generation process.
2.	Although the paper compares several causal discovery methods, it lacks comparison with some of the latest methods based on causal inference theory, such as neural network-based causal discovery algorithms. It is suggested to include comparisons with additional advanced causal discovery methods.
3.	The RoCA method involves multiple steps of noise injection and pseudo-label generation, which may lead to substantial computational costs on large-scale datasets. It would be helpful if the authors could discuss the model's complexity in more detail to assess its feasibility in practical applications.

**Questions:**

1.	The paper uses K-means and SPICE* clustering to generate pseudo-labels and achieves promising results. Would the method's stability be maintained if other unsupervised clustering algorithms were used for pseudo-label generation?
2.	If there is class imbalance in the dataset, the generated pseudo-labels might skew toward the larger classes, potentially affecting the accuracy of causal direction detection. In such cases, would the performance be impacted? If so, how could the imbalance issue be addressed?
3.	For the poor performance on Pair0071, the authors explain it as due to the presence of confounders affecting both features and labels. However, have they attempted to analyze the reasons for RoCA not achieving the best results on the Mushroom dataset across all methods?
4.	The paper does not discuss cases with confounding variables, but have the authors considered the robustness of the current method in environments with confounders? Alternatively, how might the method be improved to enhance its robustness in the presence of confounding variables?

---

> ### Author Response · Authors · 2024-11-25
> **Response from Authors**
>
> **Q1. How can we ensure that the pseudo-labels generated by clustering effectively represent the true distribution of the data? The paper lacks analysis or at least a clear statement of assumptions in this regard, which would strengthen the theoretical support for the method’s pseudo-label generation process.**
>
> A1. We have followed your constructive comments and added a clear statement of assumptions in the revised version. Specifically
>
> The effectiveness of pseudo-label generation through clustering depends on several assumptions and the suitability of the chosen clustering method for the given data. In our work, we employ two clustering methods: K-means and a self-supervised clustering method SPICE\*, each with specific requirements and theoretical support.
> Conventional K-means clustering relies on the Low-Density Separation assumption that class or cluster boundaries pass through areas of low probability density in input space for guaranteed classification performance [2].
>
> For image data, achieving low-density separation in the raw input space can be challenging due to the high-dimensional and complex nature of such data. To address the limitations of K-means for image data, we leverage a self-supervised clustering method.
> Existing theoretical studies [2, 3, 4] show that if semantically similar points are sampled from the same latent class, the representation learned by minimizing a self-supervised loss function can guarantee classification performance. Specifically, these works define semantically similar points mathematically from different perspectives such as distributional distance [2] and constraints for data augmentation [3, 4], with analyses conducted under both finite and infinite examples.
>
>
>
>
> **References**
>
> [1]: Narayanan, H., et al. "On the relation between low density separation, spectral clustering and graph cuts." NeurIPS’06.
>
> [2]: Pollard, David. "Strong consistency of k-means clustering." The annals of statistics (1981): 135-140.
>
> [3]: Arora, S., et al. "A theoretical analysis of contrastive unsupervised representation learning." ICML’19.
>
> [4]: HaoChen, Jeff Z., et al. "Provable guarantees for self-supervised deep learning with spectral contrastive loss." NeurIPS’21.
>
> [5]: Von Kügelgen, Julius, et al. "Self-supervised learning with data augmentations provably isolates content from style." NeurIPS’21.
>
> ---
>
> **Q2. The paper uses K-means and SPICE\* clustering to generate pseudo-labels and achieves promising results. Would the method's stability be maintained if other unsupervised clustering algorithms were used for pseudo-label generation?**
>
> A2. We followed your constructive comment and conducted additional experiments on the synCausal and synAnticausal datasets and incorporated different types of label errors—Symmetry Flipping (Sym), Pair Flipping (Pair), and Instance-Dependent (Ins) Label Errors—at error levels of 10%, 20%, and 30%.
>
> Specifically, we evaluated the method using various clustering algorithms:
>
> - **Agglomerative Clustering (AC):** Recursively merges pairs of clusters in the sample data using linkage distance.
> - **Gaussian Mixture Models (GMM):** A soft clustering machine learning method used to determine the probability that each data point belongs to a given cluster.
> - **Spectral Clustering (SC):** A graph-based technique that utilizes spectral decomposition to reveal the structural properties of a graph by capturing similarities among data points.
>
> In addition, we experimented with different norms for noise injection, specifically $L_1$, $L_2$, and $L_\infty$ norms.
>
> Across all these settings, our method consistently and successfully identified causal and anticausal relationships using a significance level of $p = 0.05$. The stability of the method did not diminish when alternative unsupervised clustering algorithms were employed for pseudo-label generation. This robustness suggests that the method is not dependent on a specific clustering algorithm like K-means or SPICE*, but can generalize well across different clustering techniques.
>
> The following tables summarize our findings for both datasets.
>
>
>
> ### **Table 1: Results on synCausal Dataset**
>
> | **Error Types (Including Three Levels)** | **AC Accuracy** | **GMM Accuracy** | **SC Accuracy** |
> |------------------------------------------|-----------------|------------------|-----------------|
> | Sym | 100% | 100% | 100% |
> | Pair | 100% | 100% | 100% |
> | Ins | 100% | 100% | 100% |
> ---
>
> ### **Table 2: Results on synAnticausal Dataset**
>
> | **Error Types (Including Three Levels)** | **AC Accuracy** | **GMM Accuracy** | **SC Accuracy** |
> |------------------------------------------|-----------------|------------------|-----------------|
> | Sym | 100% | 100% | 100% |
> | Pair | 100% | 100% | 100% |
> | Ins | 100% | 100% | 100% |

---

> ### Author Response · Authors · 2024-11-25
> **Response from Authors**
>
> **Q3. Although the paper compares several causal discovery methods, it lacks comparison with some of the latest methods based on causal inference theory, such as neural network-based causal discovery algorithms. It is suggested that comparisons with additional advanced causal discovery methods be included.**
>
> A3. We followed your constructive comments and included neural network-based causal discovery methods in our analysis:
>
> - NOTEARS (NeurIPS’18): NOTEARS is a widely recognized continuous optimization-based method for learning Directed Acyclic Graphs (DAGs) that represent linear causal relationships among variables.
> - NOTEARS-MLP  (AISTATS’20): NOTEARS-MLP extends the linear framework of NOTEARS to accommodate more complex causal structures by relaxing the linearity assumption.
>
> These methods have been applied across 160 different settings, covering 16 datasets with no label errors and three types of label errors: Symmetry Flipping, Pair Flipping, and Instance-Dependent Label Errors. The error levels tested were 0%, 10%, 20%, and 30%.
> The individual results will be updated in Appendix D.4.
>
> The results indicate that both methods are sensitive to label errors, as they rely on a threshold to control the sparsity of the causal relationships among variables. Finding an effective threshold for learning causal relations across different levels of label errors is hard.
>
> The average performance metrics for the NOTEARS and NOTEARS-MLP methods on both causal and anticausal datasets are as follows.
>
> | Method       | Causal Dataset Performance | Anticausal Dataset Performance |
> |--------------|----------------------------|--------------------------------|
> | NOTEARS      | 33.1%                      | 35.0%                          |
> | NOTEARS-MLP  | 52.3%                      | 40.0%                          |
>
>
> ---
>
>
> **Q4. The RoCA method involves multiple steps of noise injection and pseudo-label generation, which may lead to substantial computational costs on large-scale datasets. It would be helpful if the authors could discuss the model's complexity in more detail to assess its feasibility in practical applications.**
>
> A4. Thank you for your thoughtful comment regarding the computational complexity of the RoCA method. The computational costs are primarily raised by multiple steps of pseudo-label generation. We have acknowledged the computational costs for multiple steps of pseudo-label generation in the "Limitations and Discussion" section of the revised version of our paper.
> Note that for datasets (e.g., Coil, G241C, Digit1) with a large number of attributes larger than 200, our method requires less than one minute to produce results. In contrast, some baseline methods (e.g., RAI, ICD, LINGAM) can take multiple hours or fail to complete entirely, as they need to discover all possible causal relations in the dataset. For large-scale image datasets, such as CIFAR-10N with 50000 examples, RoCA requires approximately 2 hours to obtain results. This is primarily due to the high dimensionality of image data and the need to compute pseudo-labels and evaluate disagreement coefficients.
>
>
>
> ---
> **Q5. If there is a class imbalance in the dataset, the generated pseudo-labels might skew toward the larger classes, potentially affecting the accuracy of causal direction detection. In such cases, would the performance be impacted? If so, how could the imbalance issue be addressed?**
>
> A5. Our method is less influenced by class imbalance. According to **Theorem 1**, there is no requirement for class probability assumptions to ensure invariant disagreements under causal settings.
> An intuitive explanation is that, on causal datasets, $P(X)$ does not contain labeling information. Therefore, regardless of changes in class probabilities, $P(X)$ remains independent of $P(Y|X)$. As a result, the clustering method essentially makes random guesses with the change of class probabilities unless the clustering method itself introduces biases—such as a tendency to favor a specific class (e.g., preferring to output the negative class).
>
>
> ---
>
> **Q6.  For the poor performance on Pair0071, the authors explain it as due to the presence of confounders affecting both features and labels. However, have they attempted to analyze the reasons for RoCA not achieving the best results on the Mushroom dataset across all methods?**
>
> A6. A potential reason could be that it only includes categorical features. These categorical features are essentially an approximation of real-valued data and inherently involve information loss. This makes it challenging for clustering methods to effectively learn the underlying patterns. Consequently, the clustering process may yield less meaningful clusters, leading to less reliable disagreement measurements and affecting RoCA's ability to make accurate causal inferences.

---

### Meta-Review · Area_Chair_e7X8 · 2024-12-07

**Metareview:**

This paper proposes a noise injection method to address the limitations of existing methods on high-dimensional data (such as images). The proposed method determines the causal direction via examining the asymmetry in data distribution. It uses unsupervised clustering to generate pseudo-labels and determines the causal direction of the dataset by observing the divergence between pseudo-labels and observed labels under varying degrees of noise injection. Both the theoretical and empirical studies provide solid results and insights, effectively distinguishing causal and anti-causal data generation processes across multiple datasets.

Most questions raised by the reviewers are for clarification of important details of the proposed methods, and some questions request for extra experiments. The authors have provided detailed rebuttal, providing sufficient clarification and extra experiments. The reviewers have either increased or retained their acceptance ratings.

Overall, all reviewers are in favor of accepting this paper. I agree with this consensus and recommend an acceptance accordingly. I encourage the authors to incorporate all the extra details (e.g., extra experiments) of the discussion into future revision of the paper.

**Additional Comments On Reviewer Discussion:**

The authors have provided an effective rebuttal that push the ratings clearly above the acceptance threshold. This is not a borderline case. The acceptance consensus is strong.

---

### Decision · Program_Chairs · 2025-01-22

Accept (Poster)